# Global scale gravity wave analysis methodology for the ESA Earth Explorer 11 candidate CAIRT

Sebastian Rhode[1], Peter Preusse[1], Jörn Ungermann[1], Inna Polichtchouk[2], Kaoru Sato[3], Shingo Watanabe[4], Manfred Ern[1], Karlheinz Nogai[1], Björn-Martin Sinnhuber[5], and Martin Riese[1]

[1]Institute of Energy and Climate Research, Stratosphere (IEK-7), Forschungszentrum Jülich, Jülich, Germany
[2]European Centre for Medium-Range Weather Forecasts (ECMWF), Reading, United Kingdom
[3]Department of Earth and Planetary Science, The University of Tokyo, Tokyo, Japan
[4]Research Institute of Global Change (RIGC) and WPI-AIMEC, Japan Agency for Marine-Earth Science and Technology (JAMSTEC), Yokohama, Japan
[5]Karlsruhe Institute of Technology, Karlsruhe, Germany

**Abstract.**

In the past, satellite climatologies of gravity waves (GWs) have initiated progress in their representation in global models. However, these could not provide the phase speed and direction distributions needed for a better understanding of the interaction between GWs and the large-scale winds directly. The ESA Earth Explorer 11 candidate CAIRT could provide such observations. CAIRT would use a limb imaging Michelson interferometer resolving a wide spectral range allowing temperature and trace gas mixing ratios measurements. With the proposed instrument design, a vertical resolution of 1 km, an along-track sampling of 50 km, and an across-track sampling of 25 km in a 400 km wide swath will be achieved. In particular, this allows for the observation of 3-dimensional, GW-resolving temperature fields throughout the middle atmosphere.

In this work, we present the methodology for the GW analysis of CAIRT observations using a limited-volume 3D sinusoidal fit (S3D) wave analysis technique. We assess the capability of CAIRT to provide high-quality GW fields by generation synthetic satellite observations from high-resolution model data and comparing the synthetic observations to the original model fields. For the assessment, wavelength spectra, phase speed spectra, horizontal distributions, and zonal means of GW momentum flux (GWMF) are considered. The atmospheric events we use to exemplify the capabilities of CAIRT are the 2006 sudden stratospheric warming (SSW) event, the QBO in the tropics, and the mesospheric pre-conditioning phase of the 2019 SSW event.

Our findings indicate that CAIRT would provide highly reliable observations of global scale GW distributions and drag patterns but also of specific wave events and their associated wave parameters. Even under worse-than-expected noise levels of the instrument, the resulting GW measurements are highly consistent with the original model data. Furthermore, we demonstrate that the estimated GW parameters can be used for ray tracing, which physically extends the horizontal coverage of the observations beyond the orbit tracks.

# 1  Introduction

Atmospheric gravity waves (GWs) are one of the main mechanisms of momentum transport between different layers of the atmosphere. From their excitation processes like flow over orography, convection, jet instabilities, and other effects (Fritts and Alexander, 2003), they carry momentum to higher layers of the atmosphere by wave propagation. The GWs release this momentum when breaking, thereby accelerating or decelerating the background wind. This atmospheric momentum transport partly drives large-scale dynamical phenomena like, e.g., the Brewer-Dobson circulation and the Quasi-Biennial Oscillation (QBO) (e.g. Holton, 1992; Fritts and Alexander, 2003; Nappo, 2012). In addition, GWs also influence the occurrence and shape of sudden stratospheric warming (SSW) events (e.g. Kidston et al., 2015; Ern et al., 2016; Song et al., 2020) by preconditioning the polar vortex. They are likely a major driving force in the recovery phase of the stratospheric vortex and the downward propagation of an elevated stratopause (Ern et al., 2016; Thurairajah and Cullens, 2022; Harvey et al., 2022, 2023).

Even modern climate models are not able to resolve the full GW spectrum due to limited spatial resolution and hence the unresolved GWs are parametrized to approximate their effects. Non-orographic and orographic GWs are considered separately due to their distinct phase speed spectra and source processes. Both types are crucial for the performance of the climate model in long-term projections. For instance, they influence the frequency of SSW events (Sigmond et al., 2023, , Fig. 18) and the general dynamics of the stratosphere (Hájková and Šácha, 2024) for the orographic component and impact the QBO frequency (?Bushell et al., 2020; Richter et al., 2020) and forecasting performance (Choi et al., 2017, 2018; Kautz et al., 2020) for the non-orographic component. A better representation of these unresolved GWs creates the need for direct observations to further understand their role in atmospheric processes.

There are many different methods of observing GWs (e.g. Preusse et al., 2008, 2009). Ground based observations include airglow imaging (e.g. Takeo et al., 2017; Pautet et al., 2021; Vargas et al., 2021), meteor radar (e.g. Fritts et al., 2010; Stober et al., 2023), MF radar (e.g. Tsuda et al., 1990; Gavrilov et al., 2000; Stober et al., 2013; Minamihara et al., 2020), and lidar (e.g. Bossert et al., 2015; Kaifler et al., 2017; Strelnikova et al., 2021; Vadas et al., 2023) techniques. In-situ sensors observe temperatures and winds and are deployed on super pressure balloons (e.g. Hertzog et al., 2008; Corcos et al., 2021), radiosondes (e.g. Geller and Gong, 2010; Guest et al., 2000; Pramitha et al., 2016), meteorological rockets (Eckermann and Vincent, 1989; Goldberg et al., 2006), and commercial and research aircraft (e.g. Nastrom et al., 1987; Dörnbrack et al., 2022). In addition, also remote sensing instruments have been deployed on research aircraft in dedicated GW campaigns (Fritts et al., 2016; Krisch et al., 2017; Rapp et al., 2021) and radiosondes are launched daily in a worldwide network. However, all of these methods have observation gaps, e.g., over the oceans, and several are operated only in dedicated measurement campaigns and are often biased to strong events due to measurement planning. Satellite missions are best suited for the long-term observation of large-scale momentum transport needed for understanding global-scale processes. This was first recognized by Fetzer and Gille (1994) and Eckermann and Preusse (1999) for infrared limb observations and by Wu and Waters (1996) for microwave sub-limb observations. In particular, CRISTA (CRyogenic Infrared Spectrometers and Telescopes for the Atmosphere), SABER (Sounding of the Atmosphere using Broadband Emission Radiometry), and HIRLDS (High Resolution Dynamics Limb Sounder) have been very successful satellite missions, whose temperature measurements allowed for deriving

parts of the global GW momentum flux (GWMF) budget (Alexander et al., 2008; Ern et al., 2011, 2018). These measurements have been a keystone for better understanding GWs and improving the representation of GWs in GCMs (e.g. Orr et al., 2010; Richter et al., 2010; Stephan et al., 2019a, also Fig. 1). In addition, global navigation satellite system (GNSS) radio occultation has been used for the measurement of GWs (Tsuda and Hocke, 2002; Hindley et al., 2015). A more detailed comparison of satellite-based GW observations is given by Wright et al. (2016). Most recently, also AIRS (Atmospheric Infrared Sounder) temperature observations (Hoffmann et al., 2014; Hindley et al., 2020; Wright et al., 2022) and AEOLUS wind observations (Banyard et al., 2021) were used for inferring global GW distributions, respectively.

Gravity wave observations greatly impacted the implementation of GWs in global models. The first large push to the field was the discovery of the QBO by ? and Reed et al. (1961). Scientists were for the first time confronted by a stable, global-scale wind system that was completely independent of geostrophic balance. Lindzen and Holton (1968) solved this puzzle by wind driving through GW breaking in wind shear zones. Observations of large-scale waves and closure of the momentum balance indicate that GWs are contributing only about half of this driving (e.g. Baldwin et al., 2001; Ern and Preusse, 2009; Alexander and Ortland, 2010). However, the general idea of GWs accelerating large-scale winds, developed by Lindzen (1981) and first implemented in a GCM by Palmer et al. (1986), has proven to be essential for global wind-systems also in other parts of the atmosphere. The next important step in the development of GW parametrizations was likewise triggered by observations: the power spectrum of vertical profiles of GW-induced winds showed a universal scaling law of $m^{-3}$ (with $m$ being the vertical wavenumber) (VanZandt, 1982; D. C. Fritts and T. E. VanZandt, 1987). This discovery led to the development of spectral GW parametrizations, which are commonly used in present-day models (Hines, 1997; Warner and McIntyre, 2001; Scinocca, 2003).

Later on, the GW pattern seen in observations over regions dominated by deep convection, orography, and jets/fronts inspired the development of source-dependent parametrizations (Fetzer and Gille, 1994; Wu and Waters, 1997; Eckermann and Preusse, 1999; McLandress et al., 2000; Preusse et al., 2001; Jiang et al., 2004b, a). Richter et al. (2010) used only such physical sources and showed that coupled with a GW parametrization for interaction with the mean wind they would be sufficient to generate a realistic representation of the global circulation. Using a standard non-orographic GW parametrization, Ern et al. (2006) inferred that this best describes the observations, if the GWs are launched in the mid-troposphere and below to account for the filtering in the wind shear zones around the tropopause. Orr et al. (2010) used this knowledge and further parameters from Ern et al. (2006) to guide the non-orographic GW parametrization in the ECMWF-IFS (European Centre for Medium-Range Weather Forecasts - Integrated Forecast System), which also is employed in forecast model of the German weather center (Deutscher Wetterdienst; DWD).

Climatologies of GWMF are usually presented in terms of zonal means or monthly-mean maps. This neglects the fact that GWs frequently occur in bursts, depending on source and propagation conditions. This expresses itself in strong variations over source regions like the southern Andes (Jiang et al., 2002), but also may lead to day-to-day changes of GWMF for a whole hemisphere by a factor of three (Preusse et al., 2014). The variability of GWs in terms of the intermittency (Hertzog et al., 2012) has been measured using superpressure balloon data, but subsequently also to satellite (Wright et al., 2013; Ern et al., 2022) and radar observations (Minamihara et al., 2020). The determined intermittency was used to set up a stochastic GW

parametrization scheme (Lott and Guez, 2013; de la Camara et al., 2014), which, for example, improved the representation of the QBO in the LMD IPSLCM6 climate model (Lott and Guez, 2013; Lott et al., 2023). It is noteworthy that the intermittency is very different for orographic (high intermittency and seasonality) and non-orographic (comparatively low intermittency) GW parametrization schemes (Kuchar et al., 2020). The intermittency, of course, can be only evaluated for data sets with a large number of continuous observations in a given region and season.

It became evident, though, that GW parametrizations neglect an important feature of GWs, i.e., that GWs in the real atmosphere spread not only vertically, as assumed in the parametrizations, but also laterally. In addition, no real vertical propagation is modeled (including group velocity and possible horizontal refraction) but a check of the saturation criteria within the column is performed. Again, observations showed that GWs from convection propagate poleward by several 10° (Jiang et al., 2004b; Ern et al., 2011; Chen et al., 2019; Forbes et al., 2022) and thereby avoid being filtered in the wind reversal between tropospheric westerlies and stratospheric easterlies in the summer midlatitudes.

At the same time, the resolution of GCM models became sufficient to allow resolving the meso- to long-scale part of the GW spectrum explicitly, and particularly GCMs aiming at the mesosphere and above are striving for this solution rather than dealing with the many neglected processes in a GW parametrization, foremost lateral propagation and middle-atmosphere sources (Watanabe et al., 2008; Sato et al., 2012; Liu et al., 2014; Siskind, 2014; Becker and Vadas, 2020). Observed zonal means of absolute values of GWMF (Ern et al., 2018) and kinetic energies (Sato et al., 2023) give confidence to this new generation of middle and upper atmosphere models. Lastly, nowadays numerical weather prediction (NWP) models can be operated on the fastest supercomputers for short global simulations with resolutions high enough to resolve almost the whole spectrum of GWs, which have the potential to couple different layers of the atmosphere. Validation with observations was performed, again on the basis of the shape of the global distribution as well as on zonal mean absolute values of GWMF (Stephan et al., 2019a, b) and AIRS satellite observations of temperature perturbations (Kruse et al., 2022). The inspiration of model development by observations and the ground truth provided is hence a success story as is illustrated by the timeline shown in Fig. 1.

However, especially the high-resolution models make it obvious that new observations are urgently needed. Though the GWs themselves are resolved in these models, their momentum flux can be highly sensitive to the modeling choices, such as the representation of deep convection (?Stephan et al., 2019a; Polichtchouk et al., 2022). For instance, model uncertainties in GWMF are even larger, before tuning the sub-grid scale dissipation scheme (Franke and Giorgetta, personal communication). Furthermore, the GW community agrees that the essential quantity to observe is the direction-resolved phase speed distribution of the momentum flux (Plougonven et al., 2020). However, current generation limb sounders have only a single, sideways-directed view and thus face errors of factors of 2-5 and cannot infer the propagation direction (Preusse et al., 2009).

In order to resolve at least some of these issues, attempts were made to combine 3 profiles and thus infer a horizontal wavevector rather than an along-track wavelength (Wang and Alexander, 2010; Faber et al., 2013; Alexander, 2015; Schmidt et al., 2016). However, this is still based on non-tomographic profile retrievals, which are prone to direction flips when combining phase differences between individual profiles and induce a strong observational filter, i.e., the reduction of the visible GW spectrum due to the viewing geometry and instrument resolution. In addition, reducing the data to triples of well-matching

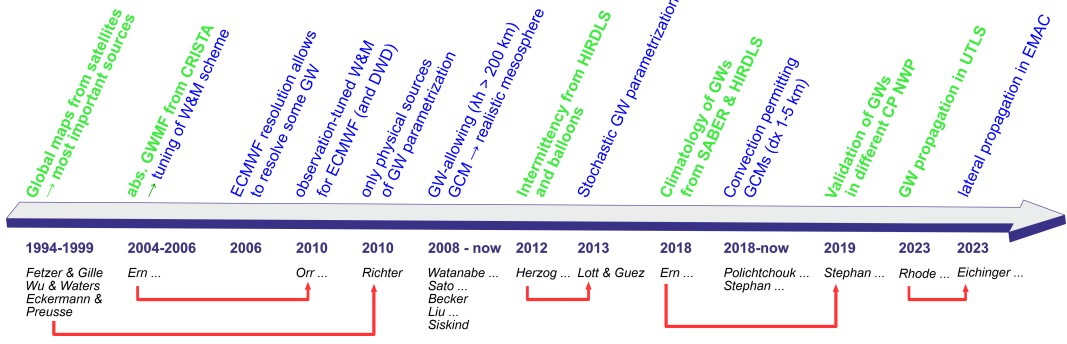

**Figure 1.** Milestones of GW observations (green) and their impact on the modeling within NWP and understanding (blue).

profiles greatly reduces the statistics by more than an order of magnitude (Schmidt et al., 2016). The only way to mitigate these problems is to observe on a dense, quasi-regular grid.

Nadir viewing instruments perform such observations. Accordingly one can infer the propagation direction (Hindley et al., 2016; Ern et al., 2017; Hindley et al., 2020) even with an accuracy that allows launching of backward ray traces from the 3D wavevectors (Perrett et al., 2021; Ern et al., 2022), but the observational filter for nadir-looking instruments is very restrictive and allows only to observe waves of very high intrinsic phase speeds (Hoffmann and Alexander, 2009; Wright et al., 2017; Krisch et al., 2020). This is problematic as a restrictive observational filter can completely distort even global distributions (Alexander, 1998; Preusse et al., 2000) and makes it impossible to see the changes in GWMF that drive the large-scale winds. In general, limb-viewing instruments, such as CAIRT, have a less restrictive observational filter for GWs.

This situation calls for an infrared limb imager in space (Riese et al., 2005). Such an instrument would now be feasible because advances in infrared detector technology made space-approved highly sensitive and fast 2D infrared detector arrays available. The general feasibility of the technique is demonstrated by the airborne GLORIA (Gimbaled Limb Observer for Radiance Imaging of the Atmosphere) instrument (Riese et al., 2014; Friedl-Vallon et al., 2014), which was deployed on two research aircraft and on stratospheric balloon flights. GLORIA has proven the ability to infer 3D temperature observations. These were used to infer 3D GW wavevectors and GWMF. In particular, ray-tracing was initialized based on these observations (Krisch et al., 2017, 2020; Geldenhuys et al., 2021; Krasauskas et al., 2023) in order to study the sources of the observed GW events and to match the GLORIA observations with ALIMA (Airborne Lidar for Middle Atmosphere Research) temperature observations taken on the same flight (Krasauskas et al., 2023). The studies demonstrate that high-accuracy wave determination is required for the scientific need and that airborne observations can deliver the necessary data of sufficient quality. In this paper, we show how the observations of a modern satellite instrument could be used to deduce global GWMF distributions and, just as importantly, individual GW parameters and events with an accuracy and detail that is unprecedented.

An opportunity to realize a limb-imaging infrared emission sounder in space is the ESA Earth Explorer 11 candidate CAIRT. In its planned design, CAIRT would be capable of measuring 3D temperature perturbations along the flight track. The envisaged altitudes of CAIRT observations range from about 5 km to 110 km; temperature observations with sufficiently low noise for

3D GW observations would reach up to ∼80 km. The spatial sampling is planned to be 50 km along the orbit track and 25 km across track over a swath width of 400 km. The limb viewing geometry allows for a high vertical resolution of 1 km.

This paper performs an end-to-end analysis of synthetic CAIRT observations. Global models capable of resolving a major part of the GW spectrum are sampled to the CAIRT observational track. Synthetic infrared observations generated by radiative transfer calculations, are perturbed by a Gaussian noise according to best estimates of actual instrument performance and are then used for a tomographic 3D retrieval to reconstruct 3D temperature fields as CAIRT would observe them. On these reconstructed fields the wave analysis is performed. The sampling along the orbit track and in particular the limited observation swath width require a dedicated wave analysis technique. In this study, we use the limited-volume 3D sinusoidal fit (S3D) method presented in Lehmann et al. (2012) for the extraction of explicit GW parameters from the measured temperatures. For any wave-analysis technique, the result depends on the configuration and setup, and thus, part of this paper is dedicated to detailing a suitable set of parameters for the estimation of GWs modeled in a high-resolution GCM simulation.

The aim of this paper is two-fold:

1. By comparing CAIRT-observed GWMF to wind-based GWMF from the full model fields, we assess the accuracy of the CAIRT observations. This follows largely the approach also used by Chen et al. (2022). Due to the high vertical resolution of the measurements, also the resulting GW drag can be estimated from the GWMF distributions.

2. By considering the interaction of GWs with the background wind, we demonstrate that CAIRT observations would open new venues for GW research.

To this end, the SSW cases of 2006 and 2019 are considered. In an SSW, GWs play a crucial role in preconditioning the polar vortex in the mesosphere. In addition, we consider the interaction with the QBO and present phase-speed spectra. Spectra derived from the synthetic satellite observations are compared with a reference from the full model. To achieve consistent results, the GW parameters must be known with high accuracy; therefore, a discussion on the impact of instrument noise on these spectra is presented. Finally, the estimated GW parameters can be used to initialize ray-tracing experiments from the orbit track to enhance and cross-validate our findings using a different, more physical method for the calculation of GW drag.

This paper begins with an overview of the high-resolution GCM data and simulated CAIRT retrievals used for the assessment. The following Sec. 3 provides an overview of the S3D wave analysis technique and the parameters used for retrieving the full spectrum of GWs from the various data sets. The adaptations of this methodology to account for the limited spatial extent along the orbits and how GW drag can be estimated from the S3D data are also discussed here. The results of the wave analysis are shown in Sec. 4, where, in particular, the GWMF and wave parameters detected in the GCM data are compared against the results from the sampled and simulated retrieved orbits of the satellite instrument. This comparison is shown for multiple scenarios of instrument noise. Afterward, in Sec. 5, case studies of the 2006 SSW event, the tropical QBO, and mesospheric GWs are presented as observed with CAIRT. A brief outlook on how ray tracing from the observed wave parameters can be utilized for supplementing the measurements is given in Sec. 6 before this study is wrapped up with concluding remarks in Sec. 7.

## 2 Data sets, GW content, and synthetic retrieval

For a comprehensive overview and validation of our GW analysis methodology, we use multiple model data sets. In addition, based on model data, CAIRT measurements are simulated by performing a simulated retrieval, which gives an estimate of a potential observation of the given atmospheric situation.

### 2.1 ECMWF IFS model data

The bulk of the simulated observations and following analysis, as well as the comparison to model data, is based on the Integrated Forecast System (IFS) of the European Centre for Medium-Range Weather Forecasts (ECMWF). In particular, we use a special model run with a horizontal resolution of TCo2559, which corresponds to about 4.4 km horizontal spacing at the equator. Every day of January 2006 the IFS is initialized by the atmospheric state of 00 UTC taken from ERA-5 and integrated for an 18 hrs spin-up period before the model states after 18 hrs, 24 hrs, 30 hrs, and 36 hrs are saved. The spin-up time allows for a self-consistent state of short- and mesoscale GWs. This model setup has a much higher horizontal resolution than ERA-5 and operational IFS analysis (e.g. Schroeder et al., 2009), which is required for this study in order to probe the short-wavelength limit of CAIRT.

The simulations have 137 vertical levels including a sponge layer of reduced strength compared to the standard IFS setup in order to limit the damping of GWs in the upper levels. The sponge layer starts above 0.78 hPa (about 50 km). The ECMWF data are generated on vertical hybrid levels. The data are then interpolated via a cubic spline interpolation to a geometric altitude grid of 500 m spacing. This undersamples the data in the UTLS but provides an oversampling around the stratopause.

In the real world, the temperature patterns along the CAIRT observation track would evolve continuously in time. For technical reasons and consistency with the model data, however, we use daily snapshots in time for the simulated observations. This leads to the observations of different orbits being simultaneous, which limits the possibility of analyzing the temperature field in the space-time domain. An analysis of planetary-scale waves like Kelvin or Rossby waves is usually the first step in a global GW analysis (e.g. Fetzer and Gille, 1994; Ern et al., 2018). Strube et al. (2020) have shown that by removing planetary-scale waves with zonal wavenumbers of up to 6, GWs can be isolated in the stratosphere and mesosphere, but close to the tropopause higher wavenumbers are needed. The GW content and the large-scale background are separated by applying the spatial methodology presented in Strube et al. (2021) directly on the model data. This scale separation approach is based on a zonal FFT with cutoff at zonal wavenumber 24 and a 3rd order Savitzky-Golay filter of 10° width in the meridional direction.

The smallest scales correctly represented by a model are typically of the order of 8 times the horizontal sampling distance (Skamarock, 2004; Preusse et al., 2014). Shorter scale fluctuations are damped for numerical stability. In case of our TCo2559 simulation, the shortest GWs resolved by the model are of about 40 km horizontal scale. Following Skamarock (2004) we have confirmed this resolution.

## 2.2 JAGUAR model data

The vertical resolution is just as important for resolving GWs as the horizontal grid spacing. Most numerical weather prediction (NWP) models, like the ECMWF IFS, focus on the troposphere and UTLS region with reduced resolution in the stratosphere and beyond. For the investigation of mesospheric GWs, a model like the ECMWF-IFS is therefore not suitable.

For testing the CAIRT GW retrieval up to the mesopause we therefore have to use different models. We require a GW permitting resolution (at least GWs of horizontal wavelengths as short as 200 km should be resolved), a high model top and dense vertical layering up to the lower thermosphere, and no sponge layer below 100 km. Further, we require the ability to nudge the model in the lower atmosphere or to assimilate other data in the mesosphere, in order to produce realistic simulations of specific situations. One model that fulfills these conditions is the JAGUAR (Japanese Atmospheric General circulation model for Upper Atmosphere Research, Watanabe and Miyahara, 2009), which is used for this study. The specific data is based on a four-day free model run initialized on 17 December 2018 at 00:00 UTC from JAGUAR data assimilation with model output every 24 hours (Okui et al., 2021). The horizontal resolution is T639, which corresponds to a horizontal sampling of $0.1875°$, or about 20 km at the equator in longitude and latitude. In the vertical, the model provides 340 pressure levels from the surface to about 150 km altitude with a sampling of roughly 300 m in the stratosphere and mesosphere. In the following, we consider the single snapshot on 18 December 2018 at 00:00 UTC.

## 2.3 Orbit data and synthetic retrieval

The GCM data provide a consistent, close-to-reality data set of global GW fields up to short scales (e.g. Stephan et al., 2019a). In order to assess the capabilities of the satellite instrument, the effects of sampling, radiative transfer, and retrieval need to be simulated.

The considered model data is given on regular grids while the limb imager will measure the atmosphere along the satellite orbit tracks. Along the orbits, a "retrieval" grid is defined by the horizontal position of the tangent points at 30 km altitude and a vertical spacing of 1 km, which will be the grid for the retrieval. Note that for all other altitudes, this deviates from the tangent point grid which is slanted, as tangent points at larger altitudes have shorter lines of sight and are hence closer to the instrument. Since the instrument measures with constant acquisition time, in a full orbit simulation, the along-track distance between subsequent observations varies between 50.2 km and 51.0 km, due to a non-circular orbit that follows the oblateness of the Earth. As an approximation, the model data is interpolated to the "retrieval" grid using a spline interpolation. Note that the resolution of the model data is finer than the observation sampling grid, and thus, the interpolation error will be small. A spline interpolation has been used because a simple linear interpolation tends to reduce GW amplitudes, which shall be avoided for a realistic comparison between simulated observations and original model data. Further note, that the retrieval will strongly smooth the temperature in along-track direction. The pixel width in across-track direction, however, is neglected.

In the next step, an end-to-end simulation of forward radiance calculation, application of noise to the simulated data, and subsequent tomographic retrieval is performed. The second edition Juelich Rapid Spectral Simulation Code (JURASSIC2; Baumeister and Hoffmann, 2022) is used to generate simulated measurements using the emissivity growth approximation

method using tabulated data (Weinreb and Neuendorffer, 1973; Gordley and Russell, 1981). Noise is applied to the simulated radiance values in accordance with the specifications defined in the Mission Assumptions and Technical Requirements (MATER). Several noise values are tested in order to specify goal and threshold values. To these synthetic radiance data, a full non-linear tomographic retrieval based on JURASSIC2 is applied (Hoffmann et al., 2008; Ungermann et al., 2010, 2011; **?**). To reduce the required computation time, JURASSIC uses a simplified setup compared to full-spectrum studies and calculates averaged radiances from precalculated tables. In this study, four spectral bands are used to determine the observed temperature.

These spectral bands capture the strong emissions of three $CO_2$ Q-branches (719.0 cm$^{-1}$ – 721.0 cm$^{-1}$, 741.0 cm$^{-1}$ – 741.8 cm$^{-1}$, and 791.0 cm$^{-1}$ – 792.4 cm$^{-1}$) as well as an atmospheric window (831.0 cm$^{-1}$ – 832.0 cm$^{-1}$) to provide a background value. In a similar form, a reduced number of spectral bands has been used before in the temperature retrieval of the CRISTA instrument (Riese et al., 1997, 1999). Trace gas volume mixing ratios of $O_3$, $CO_2$, and $H_2O$ used in the simulations are kept identical between forward simulations and retrievals.

The retrievals make use of an adjoined Jacobian computation (Lotz et al., 2012) to facilitate a conjugate gradient-based Newton-type trust region method (Ungermann, 2013) to identify the temperature and extinction field best fitting the simulated measurements. In particular, the retrievals are performed as 2-dimensional, fully non-linear tomographic retrieval slices along the orbit track. To generate the full 3D field, these 2D slices are stacked. Here, we tomographically processed half-orbits with a state vector of $\approx$100k entries and $\approx$200k radiance measurements. One such half-orbit retrieval consumes between 5 and 10 minutes, depending on convergence speed, and 10 GiB of memory on a 24-core computer. Processing a full day of data in this fashion requires $\approx$8 hours on our small cluster with 192 cores.

## 3   Wave Analysis

The main variable representing GW activity and strength throughout this study is the vertical flux of horizontal pseudomomentum, or in short the GW momentum flux (GWMF). The estimated GWMF is compared between four different data products:

D1)  zonal means of fluctuations from model data: $(\langle u'w' \rangle, \langle v'w' \rangle)$,

D2)  S3D analyses of the full model fields,

D3)  S3D analyses of the model data on the instrument sampling grid,

D4)  S3D analyses of the retrieved data on the instrument sampling grid.

The first product is calculated from the zonal $(u')$, meridional $(v')$, and vertical $(w')$ wind fluctuations and provides a true reference (at least for the zonal GWMF component, see Sec. 3.1). Note that this analysis is limited by the assumption that the background atmosphere allows for the WKB limit (Fritts and Alexander, 2003). The second product (once validated against D1) offers the spectral distribution of the original GCM data in addition to the GWMF. The third product shows the influence of the measurement grid, both in terms of the limited coverage due to the finite CAIRT orbital swath width as well as due to

the reduced horizontal sampling compared to the original GCM data (25 km × 50 km for CAIRT vs. 6 arc-minute, or about 10 km at the equator, for the NWP data).

Note that the fluctuation analysis of D1) does not contain the factor $(1 - f^2/\omega^2)$, which converts momentum flux to pseudomomentum flux, as the intrinsic frequency of the GW, $\omega$, is unknown. Here, $f$ is the Coriolis parameter at the GWs latitude. For comparisons, therefore, the S3D analyses have to be converted into momentum flux rather than pseudomomentum flux (Fritts and Alexander, 2003).

### 3.1 Fluctuations from the NWP

According to the Eliassen-Palm flux, the vertical flux of horizontal momentum of a GW is defined as the average over a full wavelength or a full wave period (or any multiple thereof) of the product of horizontal and vertical wind perturbation:

$$(F_x, F_y) = \langle \rho \rangle \left( \langle u'w' \rangle, \langle v'w' \rangle \right). \tag{1}$$

Here $F_x$ and $F_y$ are the zonal and meridional GW momentum fluxes, respectively, $\rho$ is the atmospheric density, $\boldsymbol{u}' = (u', v', w')$ is the 3-dimensional vector of wind perturbations induced by the GWs, and $\langle \ldots \rangle$ denotes the zonal mean of the bracketed quantity. Due to periodicity along a latitude circle, the zonal mean always covers a multiple of the wavelength of any wave in zonal direction, and hence, the zonal mean of the zonal momentum flux, $F_x$, can be used as a valid reference for the comparison. On the other hand, note that in the case of strong local wave events pointing predominantly in the meridional direction, the zonal mean of the meridional momentum flux, $F_y$, shows oscillations and can not be taken as a fully valid comparison reference.

For GWs, the pseudo-momentum flux is the quantity to consider for the interaction with the mean flow (Fritts and Alexander, 2003). Testing showed that the difference between pseudo-momentum flux and momentum flux is on average of the order of 20-30% for GWs larger than 100 km. For this reason, we will use the momentum fluxes, $(F_x, F_y)$, only for the validation of zonal means. Otherwise, we will use the pseudo-momentum flux throughout this article.

### 3.2 S3D Analysis of the NWP data

Compared to the analyses of Lehmann et al. (2012) and Preusse et al. (2014) a major challenge of the data used in this study is the large range of horizontal scales contained within them. As discussed in Sec. 2.1, the shortest horizontal wavelengths are about 40 km and the longest wavelengths, which occur in the tropics and subtropics, are roughly 2000 km. Thus, the resolved GW spectrum spans a range of a factor of 40 in horizontal wavelength, or about 1.6 orders of magnitude. In contrast, the S3D with a single cube size provides good results for wavelengths between half the cube diameter and three times the cube diameter, i.e., a factor of 6, or 0.8 orders of magnitude. To capture the full range of GWs, we here use a cascade of decreasing cube sizes in analogy to a wavelet analysis. The respective cube sizes for the different wave components (WCs) are given in Tab. 1.

For each cuboid size (except for the first), amplitudes and phases of all previously determined WC are refitted within the current cuboid and subtracted from the fitting volume. This avoids sampling the same wave vector multiple times with different cube sizes.

**Table 1.** Specification of the cuboid-size cascade applied to the ECMWF IFS data (lon × lat × height).

| wave component (WC) | cuboid size [km$^3$] |
|:---:|:---:|
| 0 | 500 × 500 × 10 |
| 1 | 250 × 250 × 10 |
| 2 | 200 × 200 × 5 |
| 3 | 80 × 80 × 10 |

To get reliable values for the GWMF, only the wave vectors, $k$, as determined by the cascade are used. The amplitudes of the WCs are refitted with a fixed cuboid size of 110 km × 110 km × 3 km. The GWMF values, which are directly linked to the temperature amplitudes (mid-frequency approximation, Ern et al., 2004), are therefore representative of the volume of the refit. In particular, the vertically smaller cube size of 3 km for the amplitude refit allows a better representation of strong vertical gradients of GWMF in shear zones of the background winds.

For NWP data, the refits can be performed on both temperature and vertical wind fluctuations. The ratios and phase differences between these fits can be compared to the theoretical expectations arising from the polarization relations (e.g. Fritts and Alexander, 2003) and give an idea of whether a GW is fitted or some other structure (for instance noise, imperfect background removal or other true atmospheric phenomena; cf. Strube et al. (2020)). Such unwanted structures can be rejected by allowing only deviations to the theoretical prediction up to a given threshold. Here, the wave was selected to be valid if the fraction of temperature and wind amplitude deviates from the theoretical value by a factor between 0.3 and 2.0 and the phase mismatch compared to theory was below 60°. Furthermore, we reject all wave fits with a horizontal wavelength larger than 7 times the horizontal cuboid diameter and 4 times the vertical cuboid height. Note that these are larger than previously applied, i.e., less conservative threshold parameters. These parameters, however, still lead to reasonable wavelength spectra as will be seen below.

The resulting total GWMF as estimated from the S3D cuboid on the ECMWF IFS at 25 km altitude on 3rd January 2006 is shown in Fig. 2. Shown is the sum of all four fits for the different cuboid sizes. The distribution has almost complete global coverage, i.e., there are almost no regions (or cuboids) for which not at least one sub-size in the cascade yields a valid WC. In other words, the different WCs might show gaps if considered individually but the combination of all four WCs covers the full globe. Although the larger cuboids provide more GWMF on a global scale, locally the small cuboids dominate in some regions. An example is the GWMF maximum above Iceland, which is dominated by WC 3 (not shown). This is consistent with the GLORIA Gravity Wave EXperiment (GWEX) observations during the POLSTRACC GWEX SALSA (PGS) campaign (Krisch et al., 2017), where GWs shorter than 150 km horizontal wavelength were found in the eastern part of Iceland. Smaller scale GWs originating in the Icelandic main mountain ridges are the likely reason.

In general, the estimated GWMF distribution in Fig. 2 is as expected; with strong orographic GW activity in the northern (winter) hemisphere and convective GW activity in the southern (summer) subtropics around 20°S. Also, the magnitude of GWMF is consistent with previous observations (Hertzog et al., 2008; Ern et al., 2018) as well as, for instance, dedicated

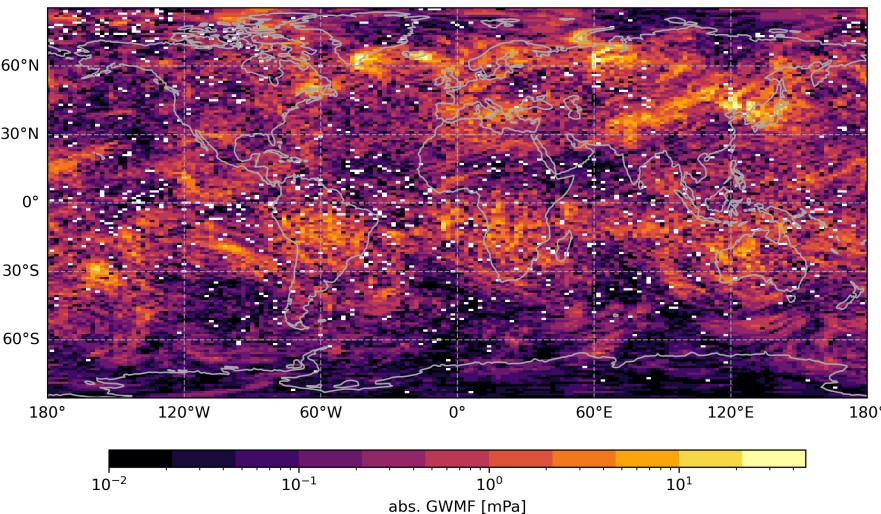

**Figure 2.** GWMF estimated from the S3D wave analysis cascade with cuboid sizes given in Tab. 1 applied to the ECMWF IFS data. Shown is the sum of all four wave components at 25 km altitude for 3 January 2006 at 12:00 UTC.

mountain wave modeling (e.g. Plougonven et al., 2013; Holt et al., 2017; Rhode et al., 2023). This distribution is therefore a
335 realistic basis for the simulated CAIRT observations.

### 3.3 S3D Analysis of sampled and retrieved orbit data

For the NWP data interpolated to the satellite tracks, the (almost) regular grid on which the S3D is performed is given by the retrieval grid. A natural specification of the cuboid size is via the number of grid points on the retrieval grid. Fig. 3 illustrates the setup and application of S3D to the orbit data.

In a similar way to the cascade for NWP data described in Sec. 3.2, a decreasing cuboid size is chosen for a total of six wave components with three different cube sizes used two times each. This doubling is done to decrease the likelihood of noise blocking a WC. The cube sizes are based on the ones shown in Sec. 3.2 but altered in order to account for the limited sampling of 50 km along track and the limited swath width of 400 km. The corresponding cuboid sizes are given in Tab. 2. The smallest cuboid size is limited by a minimum number of points to gain stable fits and, regarding cube size in km, the much coarser
sampling of the CAIRT data: applying S3D on various data sets, we found that with very few points in a cuboid, the number of direction flips, i.e., that the fitted horizontal wave vector points to the opposite direction than the reference, increases even in the absence of noise. Therefore, we cannot implement a direct counterpart to the smallest cube size chosen for the NWP data. For refit cuboid size on the orbit data, we choose $5 \times 7 \times 5$ points, or $250\,\text{km} \times 175\,\text{km} \times 5\,\text{km}$. As for the NWP data, we reject all wave fits with a horizontal wavelength larger than 7 times the respective horizontal cuboid diameter and 4 times the
vertical cuboid height of the k-fit.

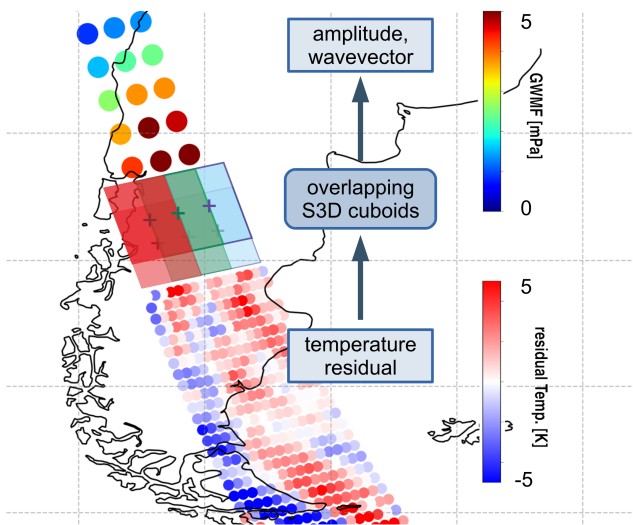

**Figure 3.** Schematics of the S3D wave analysis exemplified for an orbit segment over Patagonia. Three overlapping wave-analysis cuboids are run in parallel over the temperature residuals provided on the retrieval grid (small blue-red dots). For each cuboid center point, S3D provides the 3D wave vector, amplitude, and phase (bigger blue-green-red dots).

**Table 2.** Specification of the cuboid-size cascade applied to the CAIRT orbit data (along-track times across-track times vertical). The width of the Nyquist penalty chosen for these experiments is 0.8 in the horizontal and 1.5 in the vertical.

| wave component | cuboid size [pts] | cuboid size [km$^3$] |
|:---:|:---:|:---:|
| $0 + 1$ | $11 \times 9 \times 11$ | $550 \times 225 \times 11$ |
| $2 + 3$ | $7 \times 9 \times 11$ | $350 \times 225 \times 11$ |
| $4 + 5$ | $5 \times 7 \times 11$ | $250 \times 175 \times 11$ |

It is known that retrievals of limb instruments can generate oscillations between tangent altitudes (Preusse et al., 2002; Remsberg et al., 2008; Yue et al., 2010; Pedatella et al., 2015). In particular, these oscillations occupy the shortest resolvable vertical wavelengths. In our processing chain, this can be either mitigated by regularization in the retrieval process or, as we here choose to do, by applying a penalty function to the variational methods determining the wave vector $\boldsymbol{k} = (k, l, m)$ in the wave analysis. In particular, our S3D method uses the fitting function

$$f(\boldsymbol{x}) = A \sin(\boldsymbol{k} \cdot \boldsymbol{x}) + B \cos(\boldsymbol{k} \cdot \boldsymbol{x}) \tag{2}$$

with the spatial coordinate being $\boldsymbol{x}$. This resembles a single 3-dimensional plane wave of arbitrary phase. When searching for the best solution, we introduce an additional penalty function and minimize the sum $(f(\boldsymbol{x}) - y(\boldsymbol{x}))^2 + P(\boldsymbol{k})$, with the penalty

function $P$ as regularization to remove unwanted fits of the aforementioned retrieval artifacts. Here, $y(\boldsymbol{x})$ are the observed or modeled temperature values at the corresponding locations $\boldsymbol{x}$.

We choose the penalty function $P$ as a squared cosine function truncated at the first minimum:

$$P(\boldsymbol{k}) = \sum_{i=1}^{3} \begin{cases} 0 & |k_i| < w_i \\ h_i \cos^2 \left( \frac{\pi}{2} \frac{|k_i| - k_{\mathrm{ny}}}{w_i} \right) & else \end{cases}, \tag{3}$$

where $k_i$ are the components of the wave vector $\boldsymbol{k}$ and $k_{\mathrm{ny}}$ is the Nyquist wave number, i.e., the highest resolvable wave number. The free parameters are the penalty heights, $h_i$, and widths, $w_i$, and the normalization is chosen such that the penalty smoothly transitions from $h_i$ at the Nyquist wave number to 0 at $|k_i| = k_{\mathrm{ny}} - w_i$. Throughout our analysis, the penalty widths $w_i$ are chosen as 0.8 times the spectral sampling in all three spatial dimensions. The penalty heights, $h_i$, need to be adapted in a way that most of the fitted noise is removed from the results. Still remaining solutions close to Nyquist limit are removed in the post processing..

Although this penalty function is motivated by retrieval techniques of measurement data, we apply it to both, the sampled and the retrieved orbit data. In this way, the S3D application is identical for both data sets.

## 3.4 Drag calculations

The interaction of GWs with the background state is mostly linked to the drag exerted when GWs are breaking or saturating, which then accelerates or decelerates the background wind. This acceleration can be approximated as the vertical derivative of the GWMF (see Eq. 1) by assuming pure vertical GW propagation Fritts and Alexander (2003):

$$(X, Y) = \frac{-1}{\langle \rho \rangle} \frac{\partial}{\partial z} (F_x, F_y), \tag{4}$$

where $X$ and $Y$ are the acceleration of the background wind in zonal and meridional direction, respectively.

Since the numerical calculation of the vertical derivative amplifies the noise, the drag is calculated from a running 5 point linear fit around the target altitude. It should be noted that lateral GW propagation may induce local patterns of seeming drag: drag in the direction of the horizontal wave vector appears at the location where the wave is propagating away from and vice versa seeming drag opposite to the wave vector appears at locations where the wave is propagating towards. In general, it is expected that these local phenomena cancel for the zonal direction in the zonal mean and a general trend remains visible. However, even the zonal mean may show signs of lateral propagation in the meridional direction. In particular, this is the case for the summer hemisphere, where GWs from subtropical convective sources propagate poleward into the mid-latitude summer mesosphere/lower thermosphere (MLT). By lateral propagation, they hence avoid the critical level filtering between tropospheric westerlies and stratospheric easterlies in the summer mid-latitudes (Kalisch et al., 2014). Regions where GWs propagate into are characterized by negative "potential drag" derived from absolute values of GWMF (Ern et al., 2011), i.e.,

GWMF increasing with altitude in a strict columnar consideration - something which should not occur according to classical theory.

It is possible to address this point further by coupling a ray tracer to the observed waves and propagating them horizontally as well as upward. Based on linear wave theory, drag solely due to wave dissipation can then be calculated. This is beyond the current study, however, and only a brief outlook on the possibility of using ray tracing to enhance CAIRT measurements is given in Sec. 6.

## 4  Simulated CAIRT observations

The performance of the wave analysis on CAIRT observations is considered in terms of wavelength spectra, phase-speed spectra, and zonal means of GWMF. As reference data, the zonal mean GWMF derived via Eq. 1 is well suited. Further, the S3D analysis applied directly to the high-resolution NWP data serves as a reference for the performance of the wave analysis on the satellite orbit tracks with and without noise.

Four different noise situations are presented in this study. The one with the lowest noise level, the goal noise, refers to the performance expected for an instrument under ideal conditions. The across-track width of CAIRT depends on the number of co-added pixels of the infrared detector array. The noise values specified in the CAIRT Report for Mission Assessment (CAIRT-RfA; ESA, 2023) apply to a 50-km across-track width. For GWs, we are using an improved 25 km width (factor 2 less co-adding), and hence scale there-given goal noise values by a factor of 1.4 for goal and 2.1 for threshold estimates. The latter is used most commonly throughout the paper in order to show the expected performance. In addition, we have also scaled the CAIRT-RfA values by factors 3 and 5, corresponding to doubled and tripled goal noise, in order to justify our noise threshold.

### 4.1  GWMF spectra in terms of horizontal and vertical scales

The GW wavelength spectra for the 3rd January 2006 at 35 km altitude are shown in Fig. 4. Gravity wave spectra are traditionally provided in terms of logarithmic horizontal and vertical wavenumber based on a semi-empirical finding of separability (e.g. Tsuda et al., 2000; Ern and Preusse, 2012) and universal scaling laws. Accordingly, we bin the wavevectors from S3D into 40 equidistant bins and show the normalized sums of GWMF from all wave components. For better orientation, the wavelength values are given on the opposite axes. Figure 4a shows the spectrum of the cascade directly applied to the NWP data. A broad maximum around $(\lambda_h, \lambda_z) = (500\,\text{km}, 5\,\text{km})$ can be seen as well as local peaks around $(250\,\text{km}, 9\,\text{km})$ and $(100\,\text{km}, 7\,\text{km})$. Since this is a global spectrum, these local features can be explained by the superposition of regions with differing GW source processes and their respective scale characteristics. All scales between 50 km and 2 000 km in the horizontal and 2 km and 20 km in the vertical are well populated. The spectral distribution is, however, not homogeneous distributed in the vertical wavelengths. This is in accordance with GW physics: The decrease at short vertical wavelength is due to wave saturation and related breaking, while long vertical wavelengths are rare in the stratosphere due to associated very high intrinsic phase speeds. At higher altitudes, the maximum of the spectra shifts towards longer vertical wavelengths (not shown), which is a known fea-

ture generated by amplitude growth and wave saturation (e.g. Gardner, 1998; Warner and McIntyre, 2001; Ern et al., 2018). In general, the spectrum is compliant with our physical expectations and is similar to spectra derived by Ern and Preusse (2012).

When we compare the resulting spectrum of the S3D method applied to sampled CAIRT orbit data in Fig. 4b, we can see the effect of limitation to orbit tracks. For the most part, sampling to the CAIRT track simply means less data, and hence, worse statistics due to the gaps between the orbit tracks. In addition, the coarser sampling of the CAIRT data affects the spectra in particular at the high-wavenumber limits. The Nyquist wavelength in across-track direction is 50 km, in along-track direction 100 km. Accordingly, there is a lack of horizontal wavelengths below about 100 km. Due to the sampling distance being lower,

such short waves can only be resolved if they are propagating roughly in the across-track direction. The Nyquist limit in the vertical would be 2 km but, due to the Nyquist penalty, solutions below 2.8 km are absent (with slight accumulation at the lower edge of the spectrum).

    The subsequent panels, Fig. 4c–f, show the spectra for S3D applied to the simulated retrieval data with different noise configurations. These spectra show a pronounced signal at short vertical wavelengths. Note that this is already suppressed

by the Nyquist penalty applied during the wave analysis (see Sec. 3.3). In addition, a spurious signal around 33 km vertical wavelength and horizontal wavelengths of 50–200 km appears for the goal noise simulation. This is probably a side effect to the 2D retrieval in along-track slices combined with the S3D penalty function. With increasing noise, however, this signal is reduced. Contrary, the signal at short horizontal and vertical wavelengths becomes stronger with enhanced noise. Note again that a horizontal wavelength below 100 km is below the Nyquist limit for along-track sampling, i.e., there is evidence for a

wave-like behavior across-track despite the fact that individual tracks stem from independent retrieval slices. The distributions show that the wavelength ranges affected by the combined retrieval and analysis artifacts are located in narrow, well-defined bands. This motivates to reject all wave events with horizontal wavelengths below 100 km, all vertical wavelengths below 2.8 km and all vertical wavelengths above 33 km (the latter for the spurious part in Fig. 4c) in addition to the cut-offs applied to the individual k-fits based on the current cube-size (cf. Sec. 3.3). The noise-suppressing cut is not indicated in the spectra of

Fig. 4, but we will use it for the distributions shown in the phase-speed spectra below and in the zonal means.

    Apart from the spurious artifacts occurring at the edges of the observation limit, the general spectral shape with the maximum at $(\lambda_h, \lambda_z) = (500\,\mathrm{km}, 5\,\mathrm{km})$ is very consistent across all noise levels. If care is taken at the spectral limits, the S3D cascade is, therefore, a robust method for the derivation of GW spectra from the CAIRT instrument.

## 4.2    Zonal mean GWMF

The zonal mean GWMF calculated via Eq. 1 and from the S3D wave analyses for 3rd January 2006 at 25 km altitude are shown in Fig. 5 and 6 for zonal and meridional direction, respectively. Note that 25 km was chosen because it is the lowest altitude, where the cuboid is safely above the tropopause; hence, the reliability of the fits at this altitude is essential for the inference of tropospheric sources (cf. Strube et al., 2021). Here, we calculate the zonal mean of each wave component individually before adding them up. Thereby, we get a representative zonal mean for each wave component, ignoring locations where no

reliable fit was possible instead of setting the contribution to zero. Summing up the contributions of all wave components

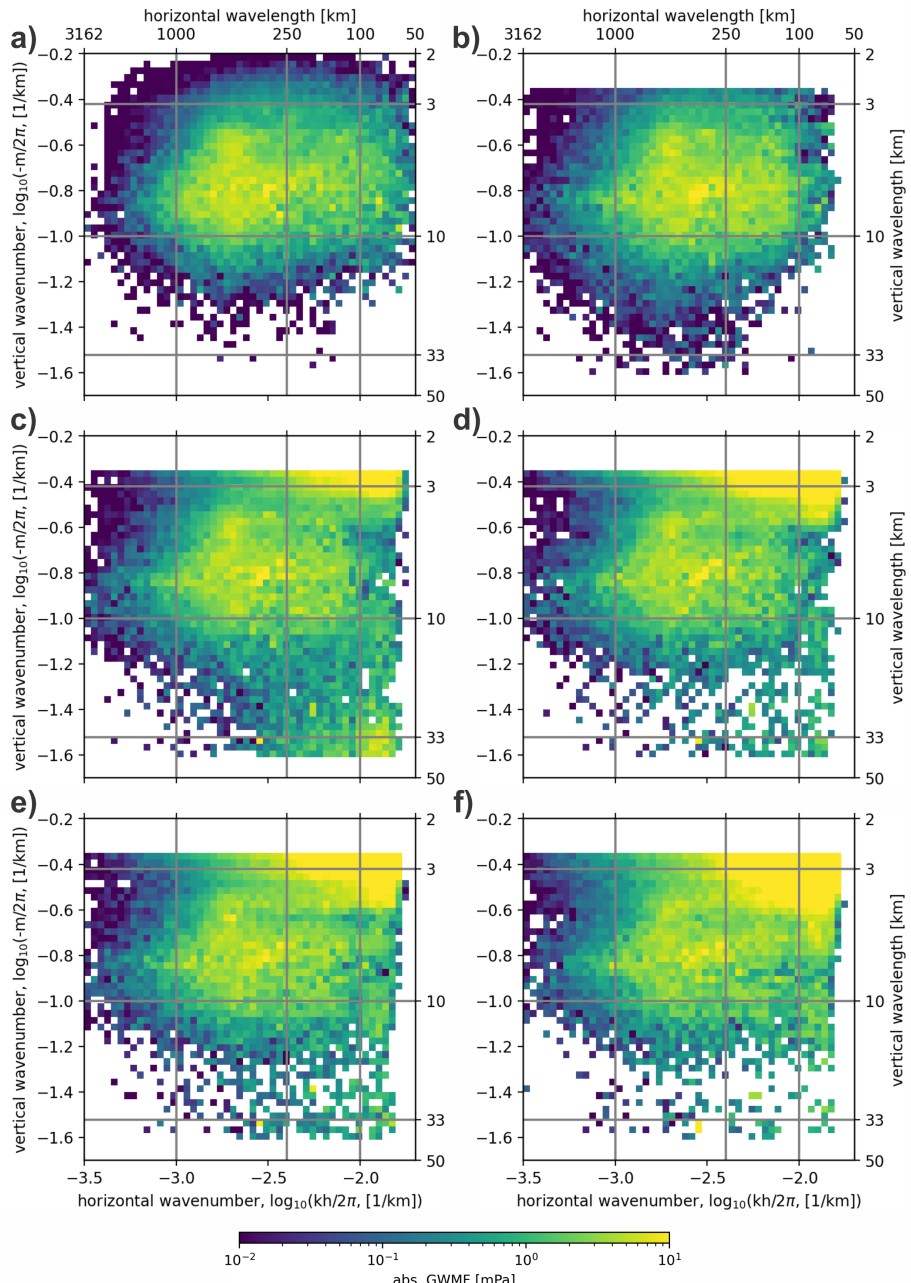

**Figure 4.** Comparison of averaged spectra for 80°S to 80°N and all longitudes for 3rd January 2006 at 25 km altitude as derived via the S3D cascade. Panel a) shows the spectrum of the NWP data on the original grid, panel b) for the orbit sampled NWP, and panels c)–f) the simulated retrievals for different noise levels of c) goal, d) threshold, e) two times goal and f) three times goal.

before calculating the zonal mean, i.e., treating non-fits as zero, leads to a reduction of the GWMF by about 20%, which gives an estimate of the uncertainty due to unreliable fits.

For the comparison, the zonal mean GWMF calculated from the wind fluctuations is smoothed by a boxcar filter of 5 km in the vertical and 100 km in the latitude direction, corresponding to the size of the refit cuboid (see Sec. 3.2). For comparability, all S3D results are converted from pseudomomentum flux to momentum flux by dividing the GWMF of each individual wave fit by the factor $(1 - f^2/\omega^2)$. The S3D analysis of the NWP-cascade data in Fig. 5a is shown separated into eastward and westward directed components as well as upward and downward propagating wave events. This distinction between up- and downward waves is only possible for the NWP data as it requires the simultaneous analysis of temperature and vertical wind perturbations. The results show that downward flux can be neglected in the ECMWF data considered here. Similarly, the eastward maximum in the southern subtropics and the westward maximum in the northern mid and high latitudes are predominantly composed of eastward and westward propagating waves, respectively. The east-west distribution will vary under different meteorological conditions. The zonal mean generated from the S3D data aligns very well with the general pattern and magnitude. Only at around 38°N, the S3D method underestimates the GWMF in comparison to the GWMF derived directly from the wind fluctuations. All in all, this shows that the S3D method is a valid tool for the estimation of GW distributions from model and satellite data.

The good matching quality is also evident in the meridional GWMF (Fig. 6a), which is separated into northward and southward directed waves. The GWMF shows notable compensation between these components, as will be demonstrated more clearly in the phase speed spectra in Sec. 4.3.

Fig. 5b shows the zonal mean after filtering of all detected GWs with horizontal wavelengths smaller than 150 km. This leads to a general reduction of the GWMF strength but also to an increased reduction in the 38°N structure. Therefore, the underestimation in this part in Fig. 5a is most likely related to GWs with smaller wavelengths than picked up by our S3D method. Note that the amplitude reduction of meridional GWMF due to the cutoff in Fig. 6b is lower than for the zonal GWMF. This indicates that the meridional flux is conveyed by longer horizontal wavelengths than the zonal GWMF.

Sampling the NWP data to the CAIRT orbit tracks shows a very comparable zonal mean. The values underestimate the GWMF inferred directly from the winds in a very similar way as the filtering of all GWs smaller than 150 km horizontal scale (Fig. 5b and 6b). At around 38°N, there is a similar stronger decrease in amplitude. This is a hint that S3D applied to orbit data misses this feature either because it consists of very short-wavelength GWs or because it is situated in a gap between the area covered by the orbits. The effect of the different noise levels and radiative transfer effects in the retrieval is shown in Fig. 5c,d and 6c,d. In general, the noise has a very limited effect on the zonal mean GWMF for both zonal and meridional direction. This has two reasons: The first is that the core of the spectrum is not affected (see Sec. 4.1). All real features in, e.g., zonal means or global distributions (discussed in the following sections) hence are contained also in the synthetic CAIRT data with retrieval noise. They may be obliterated or hidden by too strong noise, though. The second reason is that the noise does not cause waves of a preferential direction and that it accordingly averages to a zero net contribution in a sufficiently large distribution such as a zonal mean.

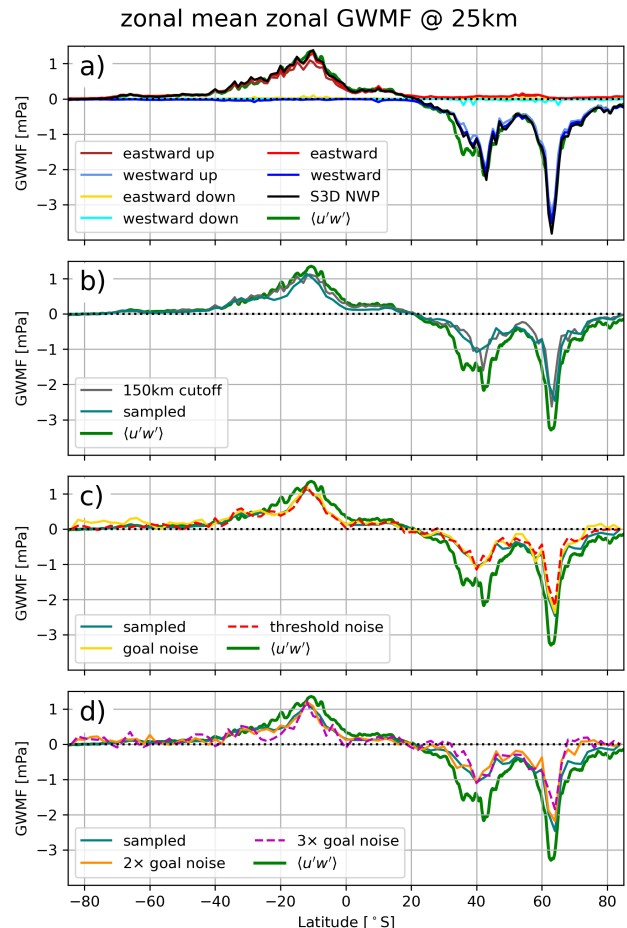

**Figure 5.** Zonal means of zonal GWMF (note: not pseudomomentum flux) from full NWP data of 3rd January 2006 analyzed with an S3D cascade (a), the same data but with a cutoff wavelength at 150 km, (b), and orbit sampled and retrieved simulation data with various noise levels (c, d). The green line in all panels shows the GWMF as calculated directly from the wind fluctuations.

## 4.3 Phase speed spectra

Phase speed distributions are essential for understanding how GW propagation and filtering by the background wind in the atmosphere evolves throughout individual events and time periods. An intuitive way of plotting them is by polar plots showing the ground-based phase speed in terms of radial distance and the propagation direction as the angle. For instance, these diagrams can be compared to blocking diagrams (Taylor et al., 1993) and indicate which parts of the phase speed spectrum are filtered by critical level filtering. Furthermore, reliable phase speed spectra are a good indication of whether ray-tracing initialized from the wave parameters will be possible. The results for an S3D cascade applied to the NWP data and simulated CAIRT

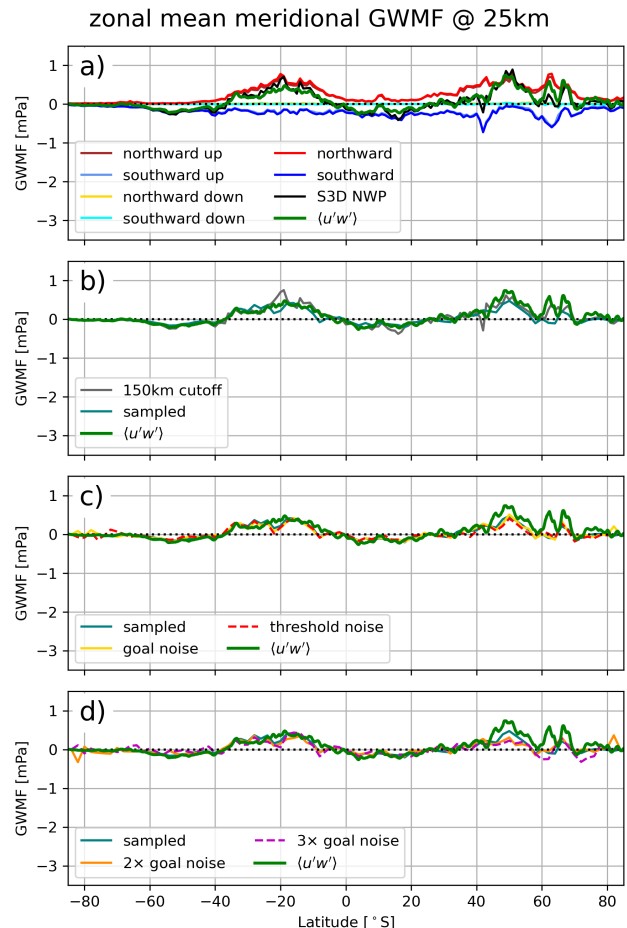

**Figure 6.** Same as Fig. 5 but showing zonal means of meridional GWMF.

observations are shown in Fig. 7. For this, the same filtering criteria as in Fig. 4 have been used. All spectra for CAIRT (panels c–f) are generated by removing the retrieval artifacts via the wavelength cuts described in Sec. 4.1.

The reference phase speed spectrum in Fig. 7a shows a superposition of two main features: north-south oriented wave events
with high phase speeds and east-west directed events with small ground-based phase speeds. While the latter are most likely of orographic origin, the former may be convectively excited. Limiting the spectrum of GWs to only waves with horizontal wavelengths longer than 150 km does not change the estimated spectrum.

Sampling of the model data to the orbit tracks and simulated retrieval with goal noise (Fig. 7c, d) show a very similar distribution. The general pattern is picked up very well demonstrating that limited coverage and coarser sampling have only a
500 minor influence. The only detriment of the spectrum can be seen in slow phase speeds in (south-)east direction, where some wave events are determined with an increased phase speed or slightly shifted direction likely due to a selection bias. With increasing noise (Fig. 7e, f), the reference pattern gets more obscured until the previously dominating north-west feature can

be hardly discerned in a background of spurious wave signatures with anomalous east-westward, or even homogeneously, distributed orientation.

In general, the good resemblance of the reference phase speed spectrum gives confidence that CAIRT would see a representative sample to estimate the full GW phase speed spectrum even in the case of higher-than-expected noise. The coarser sampling by a limited number of orbit tracks seems to be mostly irrelevant to the quantification of global phase speed distributions. This is very important for the consideration of the impact of differently generated GWs in phenomena like, e.g., the QBO and the wind reversal during an SSW.

As a comment to the zonal mean GWMF in Sec. 4.2: We see that increased noise, such as shown in Fig. 7f, generates a substantial background critical to the interpretation of the spectra, which nevertheless has little direction preference, and therefore, does not bias the zonal mean net GWMF to a larger degree (Fig. 5 and 6).

    For a more in-depth look of the effect of sampling and retrieval noise on different parts of the GW spectrum, App. A shows the phase speed spectra of the southern subtropics and the mid to high latitudes separately. For a quantification of the spectral

deterioration, we also show the circular and radial variances of the spectra there.

## 4.4 Global GWMF distribution from CAIRT

One of the advantages of space-borne measurements is of course the near-global coverage. Investigating the retrieved CAIRT GWMF on the orbit tracks gives us a first assessment of the quality of the observed GWMF distributions. A comparison of the different noise levels indicates where the noise effect is most critical and how it reduces the number of usable samples. Again,

we are investigating the 3rd January 2006 at 25 km altitude to stay consistent with the previous analysis.

    Figure 8 shows the global distributions for the detected GWMF along the satellite orbits. The sampled data presented in Fig. 8a show very good agreement with the horizontal distribution from the full NWP data (cf. Fig2). In particular, strong GWMF events are seen above the northern Atlantic and Scandinavia as well as in the southern subtropics. When performing the full radiative retrieval with goal noise (Fig. 8b), a few minor noise artifacts enter the orbits. However, these are negligible

for the general distribution of GWMF features. Once the noise reaches triple goal noise (Fig. 8d), there are spurious artifacts in all of the orbit tracks and the smaller features are much harder to discern. The most dominant features are not affected by this and are still very usable even with this high and much-overestimated noise.

    Further, we investigate the reduction in the number of usable S3D wave fits due to the increasing noise in Fig. 9. Since the dominating wave component is typically the first one, and therefore not as strongly affected by the introduction of noise, we

focus on the second wave component for this exercise. The distribution deduced from only the second wave component (Fig. 9a) shows a similar situation as the sum of all wave components (Fig. 8a). In both figures, the salient features are increased GWMF above New Foundland and Scandinavia as well as Peru, Brazil, and southern Africa. The amplitudes, however, are strongly reduced if only the second wave component is considered, which increases the likelihood of noise altering the detected wave parameters. Noteworthy, this causes not as much a background noise level but a strong reduction of the number of valid data

points resulting in gaps in the orbits. In particular, once reaching noise levels of double goal noise and beyond, regions of low

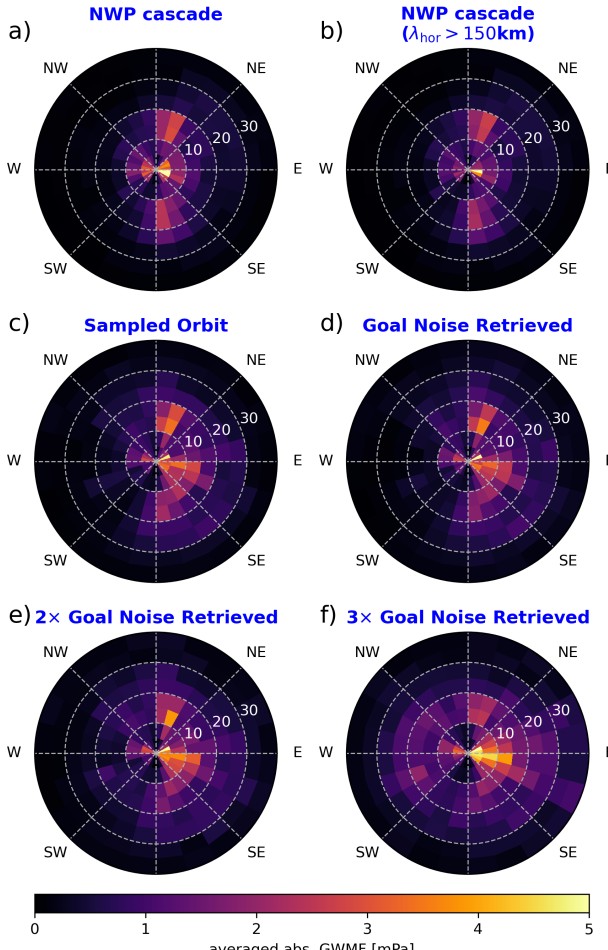

**Figure 7.** Comparison of averaged phase-speed spectra for $80°\mathrm{S}$ to $80°\mathrm{N}$ and all longitudes for NWP data of 3rd January 2006 analyzed by an S3D cascade (a), the same data but limited to horizontal wavelengths of $150\,\mathrm{km}$ and longer (b), data sampled to CAIRT grid (c), and end-to-end simulated retrievals with various noise levels (d-f).

GWMF become very sparsely sampled. In general, however, the S3D performs well in retaining the wave parameters where strong GW events are seen albeit the high noise levels.

The deterioration of the GWMF retrieval due to noise can be quantified by comparing the simulated GWMF for all retrieval setups to the results estimated from the temperature sampled directly to the orbit. This comparison is shown in Fig. 10. In the perfect case, the panels would show a straight line corresponding to unity. The deviation from this line gives a measure of the uncertainty in the GWMF estimation due to noise and can be quantified by the root-mean-square deviation (RMSD) to the reference data set. For low noise levels up to doubled goal noise, the retrieval performs reasonably well. For the sum of all WCs, most of the retrieved GWMF values are close to the center and the RMSD is around $0.5\text{-}0.6\,\mathrm{mPa}$ (Fig. 10a and b). The

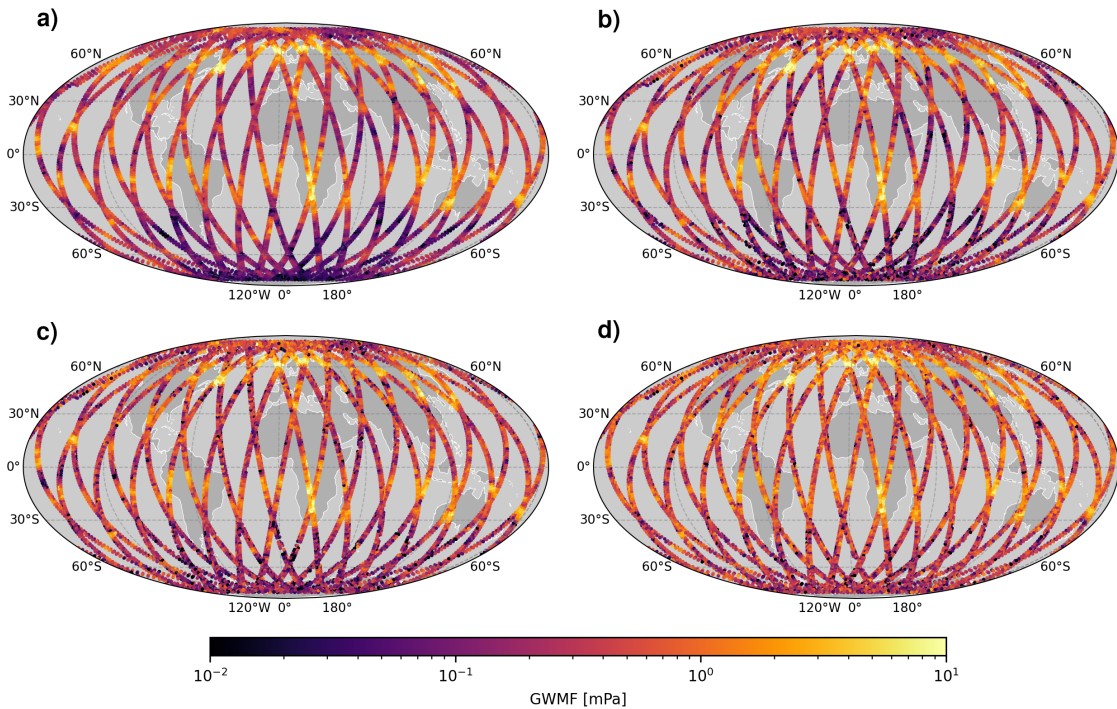

**Figure 8.** Maps of absolute GWMF summed over all WCs for a) data sampled to CAIRT orbit tracks and b)-d) end-to-end simulated retrievals with goal, double goal, and triple goal noise levels, respectively. All distributions are given at 25 km altitude on 3rd January 2006

distribution significantly widens in the triple goal noise simulation (Fig. 10c), where the RMSD increases to about 0.8 mPa.

However, a general correspondence between the reference and the retrieval-noise data is still seen, indicating that the summed GWMF distribution is still fairly usable even in the worst case scenario.

     This is not true if we look at the second wave component individually (Fig. 10d–f). The scatter distribution widens visibly with doubled noise and is almost homogeneously distributed with tripled noise. This is also reflected bey the RMSD, which increases from 0.32 mPa to 0.46 mPa. Note that these values are much more severe as in Fig. 10a–c considering the lower

GWMF detected in the second wave component.

## 5   Case Studies

To further demonstrate the capabilities of the CAIRT satellite in combination with the S3D wave analysis methodology, we investigate a few specific case studies for actual application in observing the interaction between GWs and the mean flow.

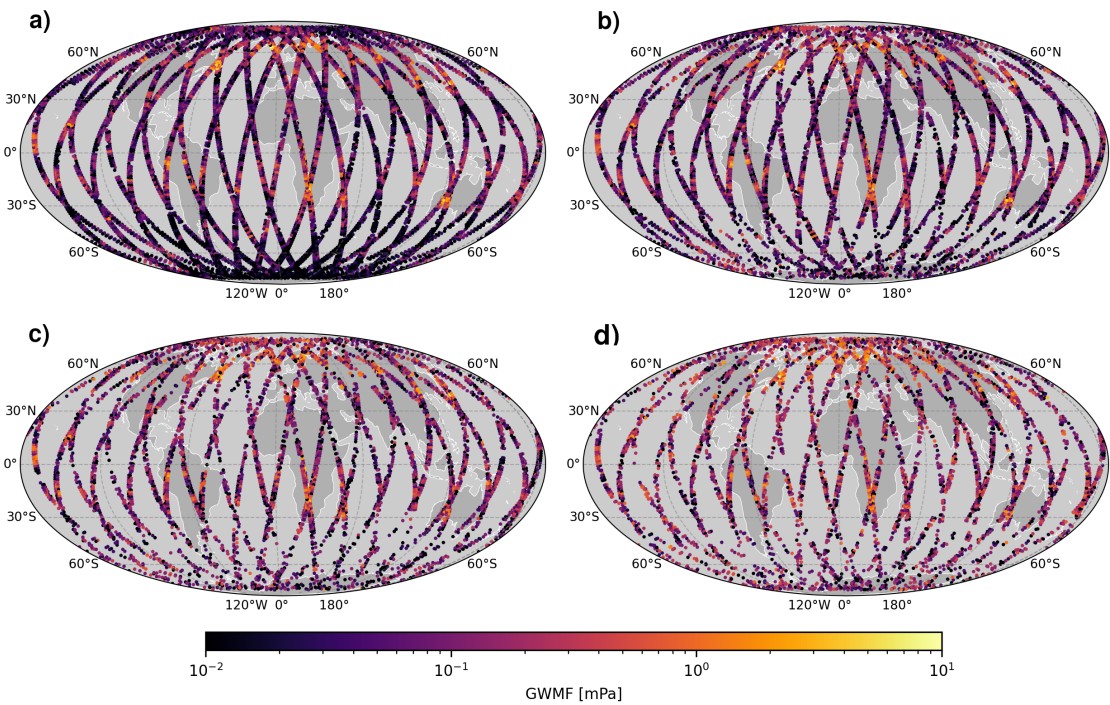

**Figure 9.** As Fig. 8 but only showing the second WC.

## 5.1 January 2006: SSW

In January 2006, an SSW occurred in the northern hemisphere: An initial displacement of the polar vortex was followed by a subsequent splitting into two weaker vortices (Ern et al., 2016; Thurairajah and Cullens, 2022). Simulated CAIRT observations allow analyzing the change of GWMF and GW drag as this event unfolds in the course of the SSW. The upper rows of Figs 11 and 13 show exemplary snapshots of the GWMF projected onto the wind direction at 25 km altitude along the orbit tracks (blue/red color scale) superimposed on the absolute wind velocity (green shading) at the same altitude, which shows the location and shifting of the vortex.

We can compare the GWMF distribution with the location of topography and other source processes, such as convection and regions of an unbalanced jet, for a first guess of the underlying excitation process. The background wind velocity is a crucial factor influencing the GWMF distribution because GWs tend to have long vertical wavelengths and can reach their largest amplitudes when propagating against strong background winds (Preusse et al., 2006). The cyclonic winds are westerly at most latitudes but, as the vortex is strongly shifted, easterly close to the pole. In general, the top rows of Figs 11 and 13 reveal enhanced GWMF values where strong wind velocity is seen. Here, negative GWMF values indicate waves propagating against the wind, i.e., potentially decelerating the mean flow. Particularly high GWMF is found above the British Isles, Norway, and Mongolia. These are all regions known to be sources of mountain waves as well as GWs excited by unbalanced jets

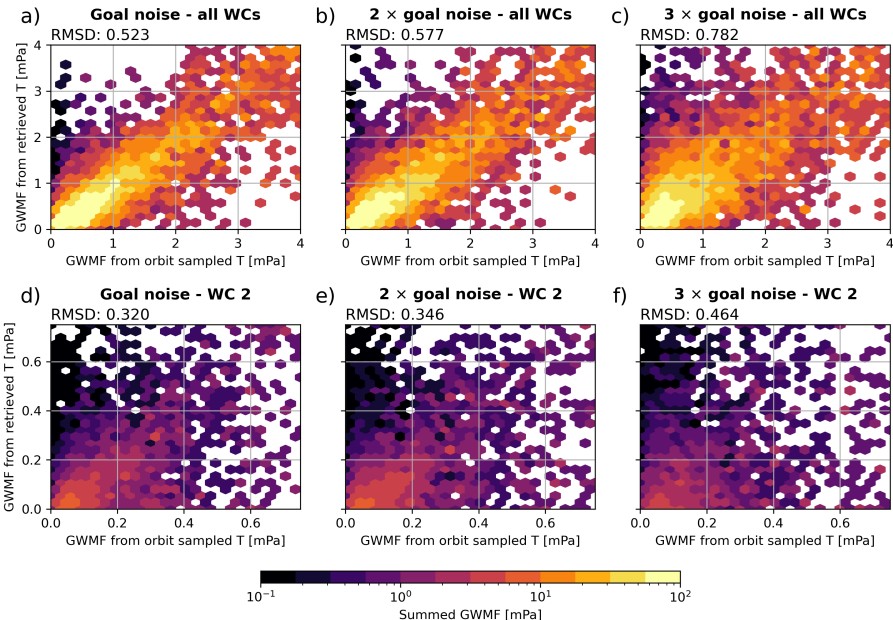

**Figure 10.** Comparison of the GWMF estimated from sampled temperature as reference (horizontal axis) and the retrieved temperatures with various noise levels (vertical axis). Top row shows the sum of all wave components (shown in Fig. 8), bottom row shows only the second wave component (shown in Fig. 9). The color shading shows the summed abs. GWMF of the respective bin - note the logarithmic color scale and the change in axis scales between rows.

(e.g. Eckermann and Preusse, 1999; Jiang et al., 2004a; Ern et al., 2018; Krisch et al., 2020). The various snapshots of the GWMF distributions show that, at 25 km altitude, the GWs have spread from their sources, which allows CAIRT to produce a representative picture of the temporal development even where the orbit tracks do not pass directly over the mountain ridges.

The rows 2 through 4 of Figs 11 and 13 show zonal means of GWMF from simulated CAIRT observations (2nd row), drag from simulated CAIRT observations (3rd row), and drag obtained from the model winds, i.e., the reference introduced in Sec. 3.1 (4th row). Note that for the generation of the latter, we have to calculate the zonal mean GWMF first before the drag can be calculated by the vertical derivative. In addition, the reference from the winds is smoothed by a boxcar filter of 3.5 km in the vertical and 2° latitude, which is comparable to the size of the cuboids from the amplitude refits.

The spatial distribution of the interplay between GWMF, winds, and drag is studied in Figure 12. At 20 km, the vortex covers a wide area with a tail extending over Japan and into the Pacific Ocean. This leads to strong GWMF over central and eastern Siberia, China, and also above the Pacific in addition to the strongest GWMF maximum above the Scandes. At 25 km and 30 km, altitude the vortex is more compact and the GWs above central and east Asia do encounter less favorable propagation conditions. Locally, drag patterns exceeding $10\,\mathrm{ms^{-1}day^{-1}}$ are forming, i.e., a sixth of the maximum wind velocity in the vortex per day. The GWs above Scandinavia are saturating and, in particular, the GWs north of the jet maximum form a second local drag maximum. The situation resembles a model experiment by Šácha et al. (2016): in a control run parametrized GWMF

was launched zonally symmetrically, in the experiment the drag of this latitude band was all focused in a box containing Japan and Kamchatka. The experimental run had a much less stable vortex and a higher tendency for SSW. Our drag maximum is further to the west (central Asia rather than the Asian east coast), but latitude and extent are very similar. Our observation is more than one week before the SSW and the local forcing may have helped to further destabilize the vortex.

Drag at 20 km altitude is much smaller than higher up and consists of opposite directions in the same pattern. Most likely we here do not observe a real acceleration of the winds but rather the effects of assuming vertical propagation in the drag calculation in regions of lateral spread of GWs from localized sources.

As the easterlies propagate downward in mid-January in Figs 11 and 13 (dashed lines around 60°N), the upper boundary of westward GWMF (red shading, 2nd row) moves downward as well, often coinciding with eastward GWMF in the easterly winds. The transition from westerlies to easterlies is connected to the shear zones and indicates wave breaking, which is then seen as GW drag (3rd and 4th row). In general, the correspondence between CAIRT-observed and reference drag is very good and captures the development in the SSW. In addition to the SSW in the northern hemisphere, the tropics show a clear crescent of positive drag wrapping around the QBO phase in both, the satellite data and the NWP. This positive drag is contributing to the downward propagation of the westerly QBO phase.

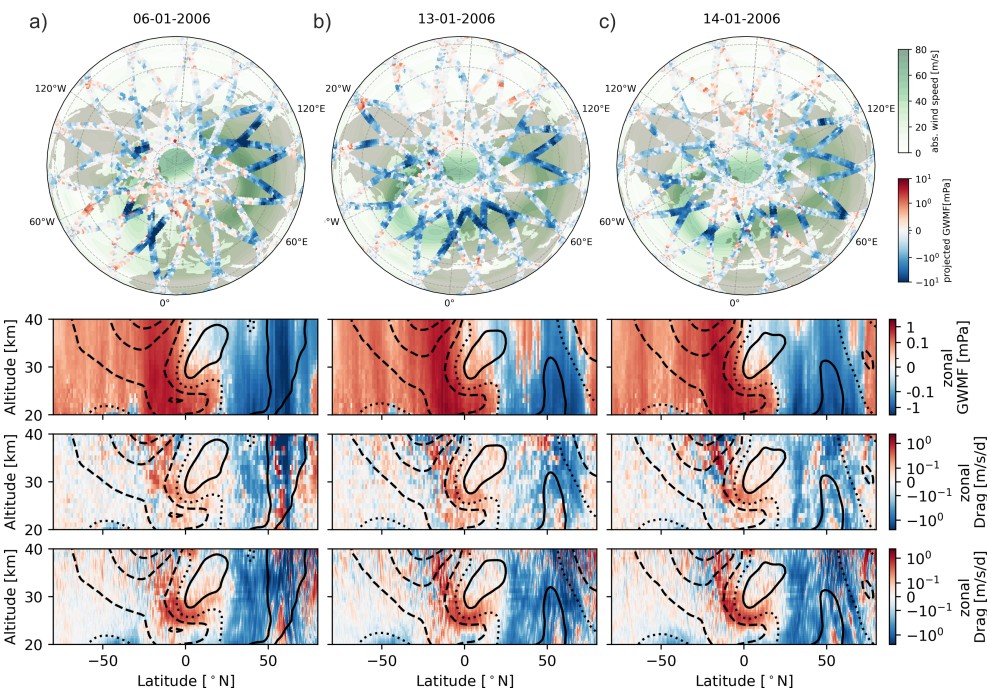

**Figure 11.** Maps of zonal GWMF at 25 km deduced from simulated CAIRT observations (upper row), corresponding zonal mean GWMF (2nd row) and drag (3rd row), and reference drag calculated from zonal mean winds (4th row). From left to right, the shown dates are 6th, 13th, and 14th January 2006. Contours of background wind in the zonal means are for $0\,ms^{-1}$ (dotted) and 5, 10, 25, and $50\,ms^{-1}$ in solid for eastward and dashed for westward winds, respectively. Note that all color scales are symmetrically logarithmic except for the wind speeds in the upper panel.

## 5.2 January 2006: QBO interaction

CAIRT would be the first instrument that allows investigation of the interaction of GWs with the QBO in detail. Missing in
previous investigations are direction and ground-based phase speed for the limb-sounding data (Ern et al., 2014) and global coverage for radiosonde data (Durre et al., 2018; Ern et al., 2023) and temporal and altitude coverage for super-pressure balloons (Vincent and Alexander, 2020). For investigating the interaction, phase speed diagrams below and above the shear zone of the QBO are inferred for 11 January 2006 as shown in Fig. 14. Contour lines of 50% and 90% critical level filtering likelihood are shown as well (see below). The inference of such phase speed diagrams is very demanding on the data quality:
they are based on the 3D wave vector, which needs to be determined with high accuracy. As seen above, they are prone to being obscured by a noise floor in case of higher simulated noise levels. As an additional challenge, the estimated wave vectors need good coverage of short vertical wavelength for the QBO: In mid-frequency approximation and in the stratosphere, where $N \approx 0.02\,\mathrm{s}^{-1}$, 3 km vertical wavelength corresponds to about $10\,\mathrm{ms}^{-1}$ (intrinsic) phase speed. A nadir-looking instrument like AIRS, which is not sensitive to GWs with vertical wavelengths shorter 12 km, i.e., $40\,\mathrm{ms}^{-1}$ intrinsic phase speed, would hence
see almost none of the physics displayed in Fig. 14.

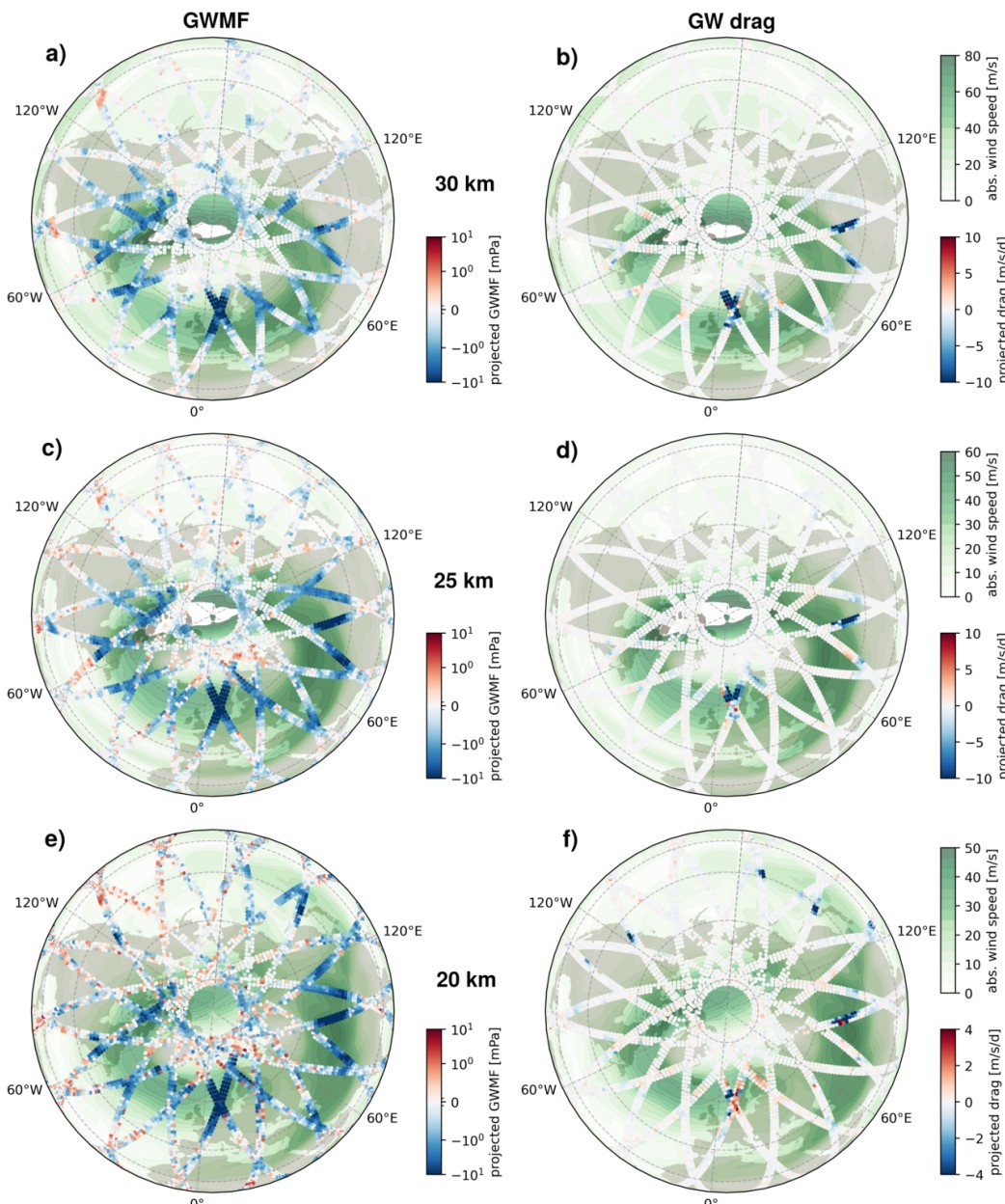

**Figure 12.** Maps of (left column) wind-projected GWMF and (right column) wind projected drag for altitudes of 20 km, 25 km and 30 km respectively. Note that GWMF uses a logarithmic color scale while drag is plotted on a linear color scale. The shown snapshot is January 11 2006 at 12:00 UTC.

The interaction of GWs with the tropical background winds at altitudes where the QBO changes from easterly to westerly phase (or vice versa) is of special interest. Upward propagating GWs break at an altitude where their ground-based phase speed

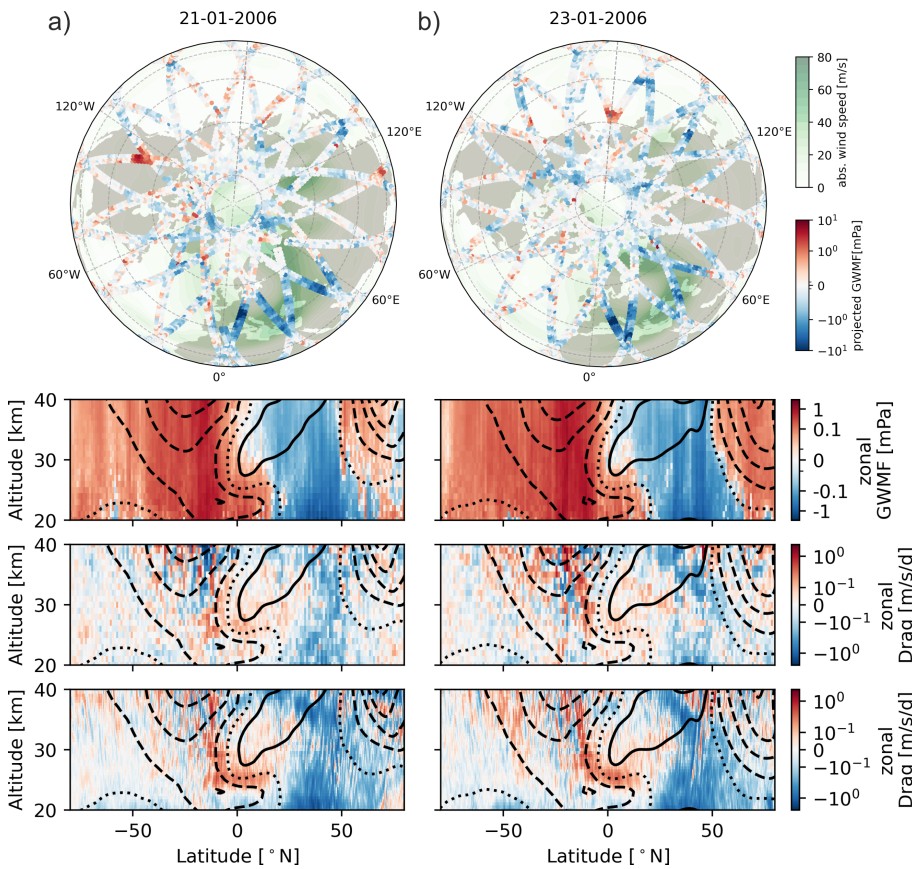

**Figure 13.** Same as Fig. 11 but for 21st and 23rd January 2006.

is the same as the background wind along the wavevector, known as the critical level. This leads to the filtering out of GWs with specific phase speeds from the full GW spectrum. The contour lines of 50% in Fig. 14 indicate phase speed regions where

critical level filtering between 8 km (the assumed maximum source altitude) and the observation altitude would occur for 50% of the tropical area (5°S–5°N). Indeed, the CAIRT simulations show a strong reduction of GWMF due to this filtering within the contour. Within the 90% contour line, the GWMF vanishes almost completely. The wind shear in the QBO region between 20 km and 25 km altitude removes almost all waves with westward ground-based phase speed. The 90% line shows two lobes, one to the northwest and one to the southwest. This indicates that GW interaction with the QBO winds is not simply a zonal

wind phenomenon as textbook examples based on Lindzen and Holton (1968) indicate. In particular, GWs at 20 km altitude show a little tail between the two lobes, where slow GWs are not completely removed but can contribute to the drag exerted between 20 km and 25 km altitude.

At 35 km altitude, as shown in Fig. 14c, the eastward-directed waves are filtered out as well. However, a few mostly north- or southward-directed GWs with high phase speeds enter the spectrum. This agrees with a reappearing of GWs in the higher

atmosphere hypothesized by Lindzen and Holton (1968). Moreover, since there are no direct sources of GWs in the stratosphere,

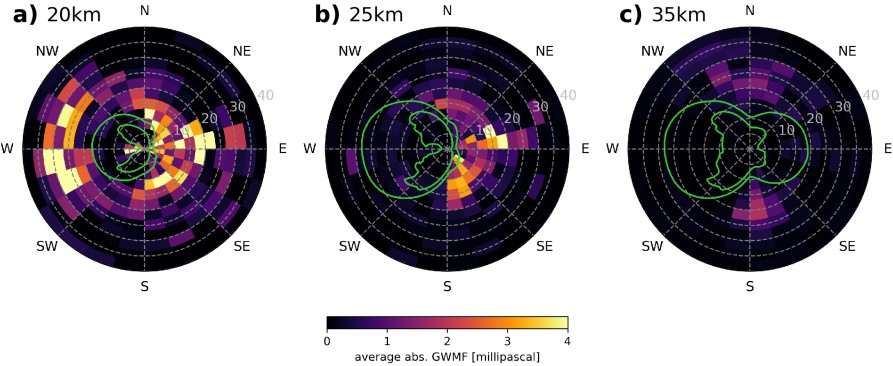

**Figure 14.** Phase speed diagrams from simulated CAIRT observations and their QBO interactions for the tropics (5°S to 5°N) at a) 20 km, b) 25 km and c) 35 km altitude. Overplotted in green are 50% and 90% blocking likelihood (outer and inner line, respectively). The analysis is performed on a snapshot of 11 January 2006.

these GWs are very likely propagating into the considered region from the subtropics, which is also seen in the more recent study of Kim et al. (2023).

The zonal mean of the resulting GW drag due to the critical level filtering is shown in Fig. 15. In the central tropics (5°S–5°N), the region considered in the phase speed diagrams, drag is exerted mainly in the shear zone indicated by the dense layering of the wind contour lines. Around 20°S–10°S, there is a region of a wider altitude range of eastward directed drag, potentially drawing the tropical jet southward. This feature can be associated with subtropical convective GW sources in the summer hemisphere, i.e., the southern hemisphere.

### 5.3 Mesospheric GWs

As known also from previous studies, the typical vertical wavelength of GWs is longer at higher altitudes due to saturation and wave breaking (e.g. Warner and McIntyre, 2001; Ern et al., 2018). For mesospheric altitudes, we thus use a larger vertical cube size, i.e., we use a vertical cube size of 11 km for cube-center altitudes of 15–34 km, 13 km for 35–54 km, and 15 km for 55–80 km.

Figure 16 shows a comparison of zonal mean zonal GWMF and zonal drag as estimated from JAGUAR data (reference) and simulated CAIRT observations throughout the whole middle atmosphere. The GWMF (reference and CAIRT) in panels a) and b) strongly decreases in shear zones and there is a notable decrease between 60 to 75 km altitude above the maximum of the winter polar vortex in the NH. Likewise, GWMF drops around the wind reversal above the summer easterly (i.e., westward) jet. GWMF also strongly decreases in the QBO shear zones, but due to the factor $1/\rho$ in the drag calculation, the drag (shown in panels c, d, and e) in the QBO is not visible with the chosen linear color scale. In the mid-latitude summer mesosphere south of 40°S, GWMF increases at higher altitudes of about 60 km but below the shear zone. This is visible as an apparent westward, i.e., negative, drag. The most likely reason is lateral propagation from subtropical convective sources (Sato et al., 2009; Ern et al., 2011; Chen et al., 2019) but alternative explanations may be the filtering of westward propagating waves. Full

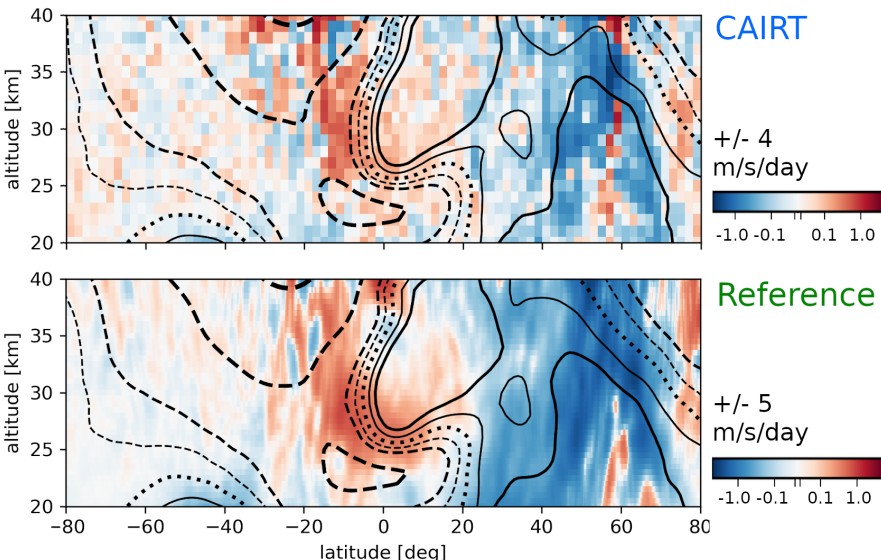

**Figure 15.** Zonal mean drag from simulated CAIRT observations (upper panel) and the model winds (lower panel), same as in Figs. 11 and 13, but with symmetrical logarithmic color scale to emphasize the QBO forcing. The linear threshold of the color scale is set to $0.02\,\mathrm{ms^{-1}d^{-1}}$. Contours of zonal background wind speed are given for $0\,\mathrm{ms^{-1}}$ (dotted) and 5, 10, 25, and $50\,\mathrm{ms^{-1}}$ in solid for eastward winds and in dashed for westward winds, respectively. As indicated above the color bars, the scales extend to $\pm\,4\,\mathrm{ms^{-1}d^{-1}}$ and $\pm\,5\,\mathrm{ms^{-1}d^{-1}}$, respectively.

wave characterization in CAIRT would provide the means to shine further light on this feature. Note that the ranges of the color scales are reduced by 25% for the CAIRT-based data. After this reduction, the patterns are very similar in values up to 70 km altitude. Above 70 km the retrieval regularization compensates enhanced noise which, however, reduces the amplitudes of the real waves, as well. Accordingly, the CAIRT-sampled data in Figure 16 show substantially higher drag above 70 km. Both negative temperature gradient in the mesosphere and decreasing density cause low emissivity at high altitudes, which limits the performance of the temperature retrieval, and therefore, 80 km is the upper limit for useful GW observations from CAIRT.

As a further example of the expected measurements in the mesosphere, Fig. 17 displays global horizontal distributions at 70 km altitude and compares absolute GWMF from the original JAGUAR data and a simulated retrieval, respectively. The shown situation is rather complex with multiple hotspots around the globe. Although the limited sampling due to the orbit tracks results in missing more detailed information on the extent of these structures, the satellite observes all major locations on the globe with compatible amplitudes. Only above South Africa, there is a slight overestimation of the total GWMF. This is caused by the orbit passing exactly over a single wave event with particularly high amplitude. In addition, regions in between the global hotspots are underrepresented in the CAIRT simulations.

More detailed assessments of the data shown in Figs. 16 and 17 is given in App. B.

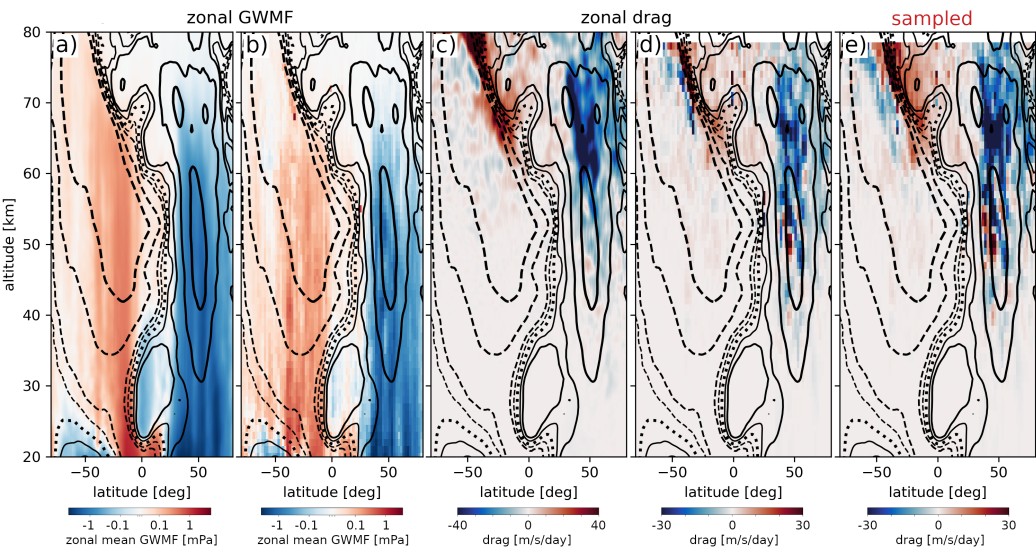

**Figure 16.** Zonal mean zonally directed GWMF derived from JAGUAR wind fluctuations (a) and from S3D analysis of the end-to-end simulated CAIRT temperature (b). Panels c)–e) show the zonal mean GW drag derived from the JAGUAR wind fluctuations (c), end-to-end simulated CAIRT temperature (d), and model data sampled to the CAIRT grid (e). Contours of the zonal mean zonal wind speed are given for $0\,\mathrm{ms}^{-1}$ (dotted) and $6.25\,\mathrm{ms}^{-1}$, $12.5\,\mathrm{ms}^{-1}$, $25\,\mathrm{ms}^{-1}$, and $50\,\mathrm{ms}^{-1}$ with dashed and solid contours for negative and positive values, respectively. The exemplary analysis period is December 18 2018.

## 6 Ray tracing outlook

One way to use the S3D wave analysis for further analysis is the implementation of ray tracing. The results of the S3D analysis provide a full characterization of the wave, and hence, can be taken as the starting points for the trajectory calculation launching one ray for each cuboid and each valid wave component. The ray tracer GROGRAT (Marks and Eckermann, 1995; Eckermann and Marks, 1997) used in this study is an updated version modified to account for the spherical geometry according to Hasha et al. (2008) and Kalisch et al. (2014). The consistency of this approach has been demonstrated by Krasauskas et al. (2023), where two independent measurements are matched via ray tracing. The synoptic-scale component of the JAGUAR model data after scale separation (see Sec. 2.1) provides the background atmosphere through which the GWs are propagated. In this section, we consider the same date as in Sec. 5.3, i.e., 18 December 2018.

For the data presented in this section, the individual ray traces are binned to a grid of overlapping grid cells with size $3° \times 3°$ in longitude and latitude sampled every $1.5°$ longitude and $1.5°$ latitude. By summing the individual contributions of all rays passing through a given grid cell, the total momentum flux or drag within this cell is calculated. The overlapping bins allow for a smoother distribution of the represented values but lead to an overestimation of GWMF at the individual location which is compensated by multiplying with a factor of 0.25.

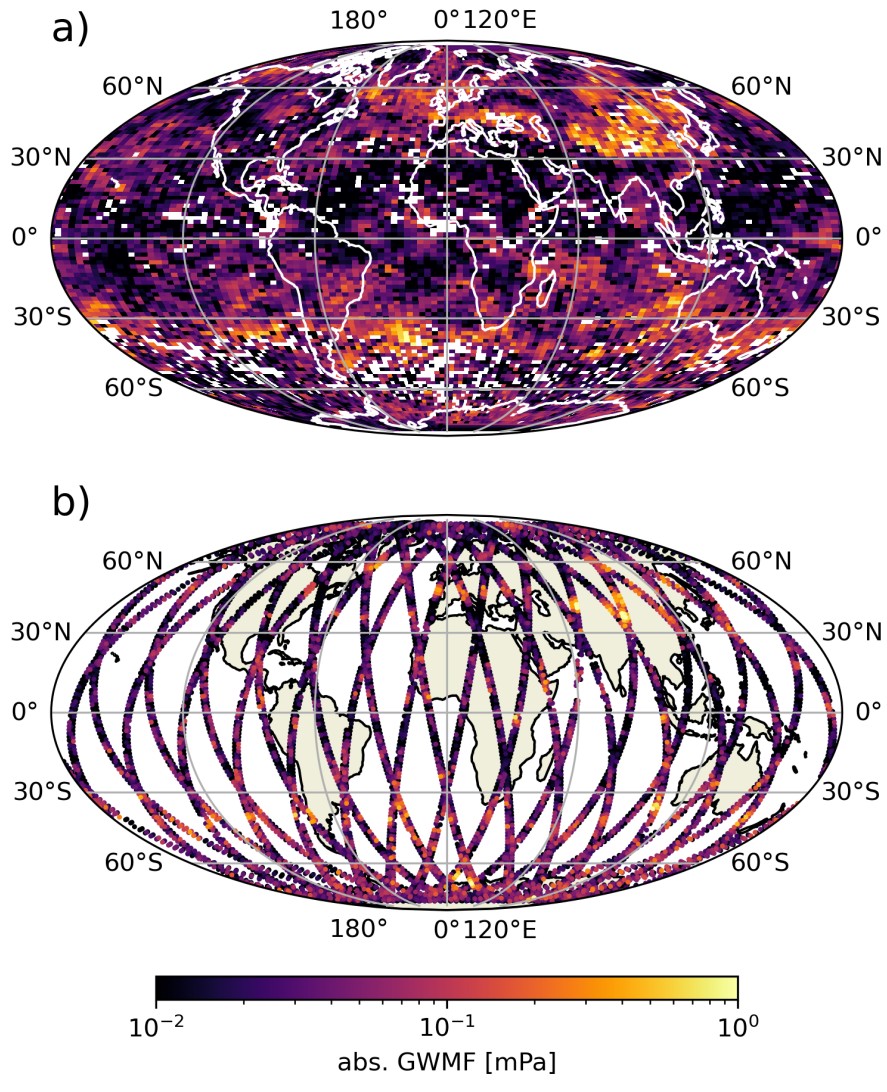

**Figure 17.** Global distribution at 70 km altitude of absolute GWMF deduced from S3D applied directly to the JAGUAR model data of December18 2018 (a) and to the simulated retrieval with threshold noise (b).

The first application of ray-tracing demonstrated in this study is spreading the GWs from their initial orbit tracks to the surrounding, i.e., we aim at a distribution where the gaps between the observation tracks are closed by the horizontal propagation of the GWs. Figure 18 shows the GWMF resulting from such a ray-tracing experiment at 25 km and 45 km (panels a and c) and the comparison to the GWMF estimated directly from JAGUAR (panels b and d). The rays are initialized from S3D analyses performed at 15, 20, 25, 30, and 35 km altitude and normalized by the number of initializations for each altitude level. Only forward, i.e., upward, ray tracing is considered here. We can see how the GWs spread from the original orbits: at 25 km the ob-

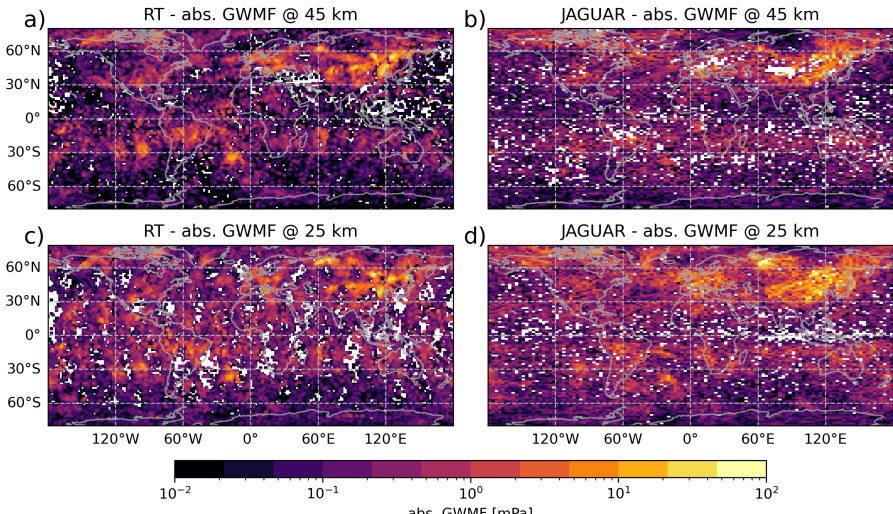

**Figure 18.** Absolute GWMF from ray tracing experiment. The rays were initialized from the S3D analyses data of 18th December 2018 00:00 UTC at 15, 20, 25, 30, and 35 km altitude from the JAGUAR simulations (see Sec. 5.3). Panels (a) and (b) show the resulting absolute GWMF at 45 km and 25 km, respectively.

servation tracks are still discernible and strongly overemphasized, while at 45 km the distribution is almost homogeneous. This methodology might therefore be used to reach an almost complete coverage by combining launch levels of different altitudes if the GW spectrum along the measurement tracks is representative of the gaps. The comparison to the S3D analysis directly applied to the JAGUAR data (Fig. 18b and d) confirms the strong bias to the orbit tracks at 25 km. This bias is alleviated to some extent at 45 km altitude and the most prominent large-scale structures are resembled within the ray-tracing simulation. Also note that the ray tracing results are better at resembling larger scale structures simply due to more and stronger GWs being present in these regions (e.g., southern subtropics, Himalaya, and Scandinavia).

Furthermore, the vertical cross-section of zonal mean zonal GWMF from the ray tracing data is shown in Fig. 19a, which can be directly compared to the model data in Fig. 16a and b. The magnitude of the GWMF agrees with the previously found distributions from S3D. At higher altitudes, the GWMF deviates from the S3D data due to no more ray initializations beyond 35 km. This leads to a mismatch at around 60°S and 60 km altitude. In general, however, the distribution is represented well, and hence, the ray tracing provides a valuable supplement to the orbit track data by allowing for a better characterization of the actual GW distributions and structures.

Furthermore, the ray tracer allows for the estimation of the GW drag for each GW along its path. In contrast to the previously shown drag estimations (see Sec. 3.4), this method does not assume strictly vertical propagation and is thus much better suited to describe the actual drag resulting from GW saturation and breaking. In this way, no artifacts due to the horizontal propagation are introduced.

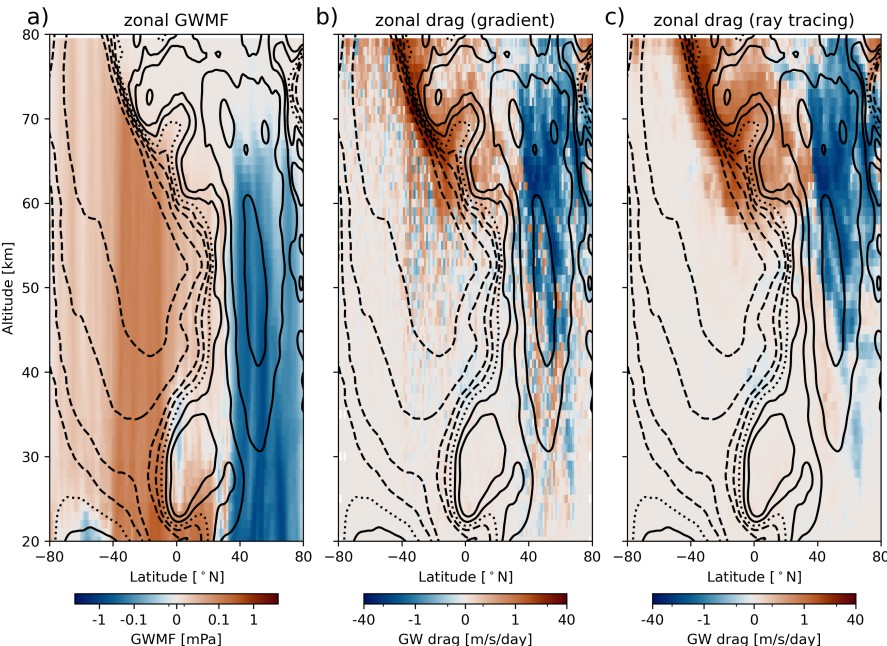

**Figure 19.** Zonal mean zonal GWMF (a) and drag (b and c) as estimated from the ray tracing experiment. Panel b) and c) show the GW drag estimated as the vertical derivative of the GWMF distribution and directly taken from the ray tracer, respectively. The situation and wind profile are the same as in Fig. 16.

Figure 19b and c show the resulting zonal mean cross-section of zonal drag calculated as the vertical derivative and directly
taken from the ray tracer, respectively. Note that, unlike before, the GWMF profile has not been smoothed before calculating
the vertical derivative. The general shape of the zonal drag is very comparable to the one seen in Sec. 5.3 with only minor
deviations in higher altitudes. The reason for this deviation, as for the GWMF, is that the highest altitude, where rays are
initialized, is 35 km. Nevertheless, most of the drag profile can be explained by upward propagating GWs originating at lower
altitudes. In particular, we find a deceleration in the polar vortex of the winter hemisphere and above the shear zone in the
summer mesosphere (around 40°S).

Comparing the methodology of drag estimation between 19b and c, the vertical derivative shows a lot of small-scale variability, which is the reason we applied a smoothing before the drag estimation in the previous considerations. The drag from
the ray tracer is much smoother and can therefore give a drag distribution at the native vertical resolution of the instrument. A
better representation of the fine-scale structures is the result. In particular, a north-south asymmetry of negative and positive
drag is seen around the polar vortex around 35 km altitude and above as well as a much more defined positive drag between
the summer easterlies and the tropical westerlies in the mesosphere.

We have demonstrated two applications for ray-tracing: The potential to fill gaps between the observation tracks in order to
gain full global coverage (similar to reverse-domain filling methods for tracers (e.g. Schoeberl and Newman, 1995; Hegglin

et al., 2004)) and the option for a complementary method to calculate drag which is valid also under lateral propagation. A further application is the inference of sources as demonstrated by studies of NWP model data in general (Strube et al., 2021) and during the 2018 SSW (?), synthetic limb sounding data (Preusse et al., 2014), AIRS observations of the 2022 Tonga eruption (Ern et al., 2022), and also GLORIA and AIRS observations (Krisch et al., 2017; Perrett et al., 2021; Krasauskas et al., 2023). In order to identify the precise location, the wave vector needs to be identified with high accuracy (Krisch et al., 2017). Slower waves which propagate more laterally are more challenging in this respect. This is likewise true also for the forward ray-tracing presented here.

## 7 Conclusions

In the past, satellite observations of GWs helped to understand fundamental science questions concerning GW sources and the effects of GW propagation, thereby inspiring the development of GW parametrizations in global atmospheric models. However, most of the observations were based on the nadir-sounding geometry, which provides only coarse vertical resolution and thus only a minor part of the GW spectrum. Recent limb sounders improved the situation in terms of vertical resolution but could not provide the quasi-regular sampling required for a full wave characterization. An infrared limb imager in space has the potential to provide these data. The novel data would also be a huge advancement in mid-term weather forecasts and climate projections since kilometer-scale global models require validation of their resolved GWs. Further, GW parametrizations will be in future use for long-term runs and require tuning to observations. And since CAIRT observes a wide range of altitudes, the GW dissipation and propagation are observed and could be used for more advanced parametrizations of these processes. The validity of the existing parametrizations could be investigated giving an estimation of the importance of non-linear effects. In addition, also artificial intelligence-based approaches will need real-observation training data sets. There are still many open science questions related to, for instance, tropospheric and, in particular, middle-atmospheric GW sources (including secondary wave generation) and the lateral propagation of gravity waves. Though we know the general importance of such processes, quantitatively it is still unclear how GWs rebuild the mesospheric vortex after an SSW, how they reach the summertime MLT, or how they enter the thermosphere, to name just a few examples. In this study, we hence have quantitatively assessed and demonstrated the potential of an instrument based on an actual instrument specification: the CAIRT mission proposed for ESA's Earth Explorer 11.

Such a demonstration depends on available tools, which we have optimized for this study: the radiative transfer and retrieval model JURASSIC and the wave analysis method S3D were developed for analyzing data measured by the CAIRT demonstrator instrument GLORIA and have been in use for about one decade. Further optimizations to the process were carried out in the last two years. Hence, this study gives a realistic appraisal of the potential of CAIRT. Until launch, it is conceivable that improved algorithms will be developed, which further enhance the GW observation capabilities of CAIRT.

In this study, the wave analysis tools are scrutinized by comparison to reference data. The example of the S3D cascade applied to the original lon-lat NWP grid shows that GWMF can be inferred using polarization relations. Both, for NWP and CAIRT data, a cascade of different cuboid sizes is required. For CAIRT retrievals, a Nyquist penalty has to be included,

which rids the obtained wave spectrum of nonphysical and noise-like waves at the shortest theoretically resolvable scales (also, without penalty, the target range of wavelengths is not reliable; not shown).

Coarser sampling cuts into the range of the horizontal wavelengths which can be captured by the wave analysis. A reduction from 10 km sampling (ECMWF IFS NWP data) to 25 km × 50 km (CAIRT) results in the loss of most of the waves with wavelengths smaller than 100 km even before radiative transfer and retrieval are applied. The ECMWF IFS data discussed here still lack scales shorter than 40 km; however, waves shorter than 20–30 km cannot propagate freely for larger vertical distances but are reflected (e.g. Preusse et al., 2008). There is no reliable quantification on the global scale (for this we would need CAIRT) but a rough estimate is that GWs shorter and longer than 100 km contribute about half of the total GWMF each (Preusse et al., 2008, 2012; Polichtchouk and Scott, 2020).

The impact of noise is mainly seen in the different spectra. Spectral distributions with respect to horizontal and vertical wavelengths indicate that the retrieval process projects this noise in particular to the Nyquist wavelength, both vertically as well as horizontally and, interestingly, also in the across-track direction even though independent retrievals of parallel tracks were conducted. This provides an opportunity for noise reduction by spectral filtering the shortest waves and leaving the most important central part of the spectrum relatively unaffected. The phase speed spectra are affected the worst but are essential for understanding the interaction between GWs and background winds. At goal noise they are almost perfectly recovered, at twice goal noise they can still be well discerned, and at three times goal noise they are obscured by the background noise, in particular for regions of moderate GW activity.

The case of the SSW in January 2006 was considered through the eyes of CAIRT. Our findings indicate that the CAIRT sampling is sufficient for capturing the daily variations of GWMF and drag. Inside the vortex, GWs propagate primarily opposite to the prevailing winds. Strong GWMF is found in regions where both strong sources exist and the wind favors GW propagation. These sources, and also the fact that the vortex is tilted and twisted with altitude, cause strong local drag patterns, which may help to amplify the instability of the vortex and play a role in the generation of the SSW (Šácha et al., 2016).

Besides capturing the GW forcing of the QBO, CAIRT can also explain its mechanism by critical level filtering seen in the phase speed spectra. This would be a unique feature that cannot be obtained by any other observation technique currently available.

Furthermore, the direct observations of the GWs along the orbit tracks can be augmented with ray tracing by initializing rays with the wave parameters given by the S3D analysis. This provides a tool for alleviating the gaps between adjacent measurement tracks by horizontal propagation, thereby giving a comprehensive global picture of GW distribution. Secondly, the GW drag can be estimated directly from the ray tracer, which clears the assumption of vertical propagation only and reduces associated artifacts. And finally, backward ray tracing is (although not demonstrated in this study) a well-tried tool for identifying GW sources.

Beyond individual observations, the CAIRT mission might be able to estimate trends in GW activity during it's planned lifetime of 5 years (with a threshold lifetime of up to 10 years) but this is beyond the current study. The daily global coverage allows for observation of daily variability of the global GW distributions.

In summary, the proposed implementation of an infrared limb sounder by ESA's Earth Explorer will deliver an instrument with sufficient spatial resolution and coverage and sufficiently low noise levels to open new venues for GW research. It thus has the potential to inspire new, urgently needed model developments for mid-term weather forecasts and climate projections.

*Code and data availability.* The data and code can be requested from the authors.

## Appendix A: Latitude-separated phase-speed spectra

Different parts of the wave spectrum can be investigated by restraining the analysis to the regions where these predominantly occur. In this case, we separate the analysis region to the (southern) subtropics, 35°S to 5°N, where mostly non-orographic GWs are expected, and to the northern mid and high latitudes, 30°N to 85°N, where orographic GWs play a dominant role. Fig. A1 and A2 show the phase-speed spectra for both regions, respectively.

The separated parts of the spectrum show more-or-less the same results as shown in Fig. 7. The phase-speed spectra degrade in a similar way with increasing noise levels. In particular, the subtropic phase-speed spectra in Fig. A1 show good agreement with the reference up to the highest noise levels. The mid and high latitudes in Fig. A2 show, as expected, slower phase speeds but also a strong reduction in GWMF from the filtering of horizontal wavelengths shorter than 150 km. The sampling to the orbit itself introduces some artifacts at higher phase speeds and dilutes the direction of the spectrum. Considering the retrieval

noise, the spectrum only further degrades when the noise reaches three times the goal noise level.

To quantify the deterioration of the phase speed spectra, we calculated the circular variances weighted with the GWMF of the individual waves and the weighted mean phase speeds for the different situations in Fig. A3a and b, respectively.

The circular variance is calculated via:

$$Var = 1 - \frac{1}{\sqrt{\sum_i F_i}} \sqrt{\left(\sum_i F_i \cos\theta_i\right)^2 + \left(\sum_i F_i \sin\theta_i\right)^2}, \tag{A1}$$

where $\theta_i$ is the direction of the GW and $F_i$ is the GWMF carried by GW $i$. This variance is 0 if all waves point in the same direction and 1 if their pointing is uniformly distributed.

The phase speed spectra for the subtropical region shows a distinct directionality, which is also captured by the circular variance in Fig. A3 a. By sampling to the orbit data, this directionality is reduced to some extent. When retrieval noise is introduced, the direction of the spectrum is recovered well for the goal noise but almost vanishes with threefold noise, giving

a circular variance of 0.8 indicating almost no distinct pointing direction. The weighted mean phase speed, on the other hand, is strictly increasing when sampled onto the orbit. This is also visible directly in the phase-speed spectra, where the center part gets spread to higher phase speeds, indicating that the spectrum itself shifts to higher phase speeds. Introducing retrieval noise further continues this trend of reduced circular variance and increased mean phase speed. Both measures together indicate that

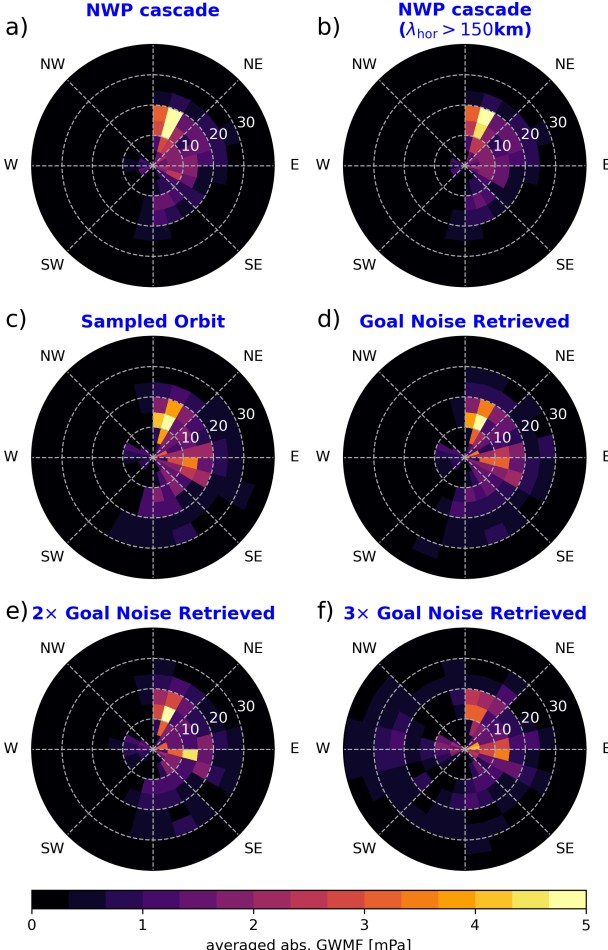

**Figure A1.** Comparison of averaged phase-speed spectra for 35°S to 5°N and all longitudes for NWP data of 3rd January 2006 analyzed by an S3D cascade (a), the same data but limited to horizontal wavelengths of 150 km and longer (b), data sampled to CAIRT grid (c), and end-to-end simulated retrievals with various noise levels (d-f).

in case of subtropical GWs, the artifacts due to retrieval noise introduced in the spectra are non-directional and mostly in the
higher phase speeds and are note easily distinguishable from the actual spectrum.

In the mid to high latitudes, the circular variance decreases strongly compared to the model simulations, where it is almost
1 due to the two dominant directions. The sampled and retrieved data are missing part of the westward GWs, leading to a
reduction in circular variance. With increasing retrieval noise, the circular variance decreases further, indicating that the noise
in this situation has a small directionality (which can also be seen in Fig. A2f). Just as in the subtropics, the weighted mean
phase speed increases with increasing noise. This effect is, however, smaller since the phase-speed spectrum already shows

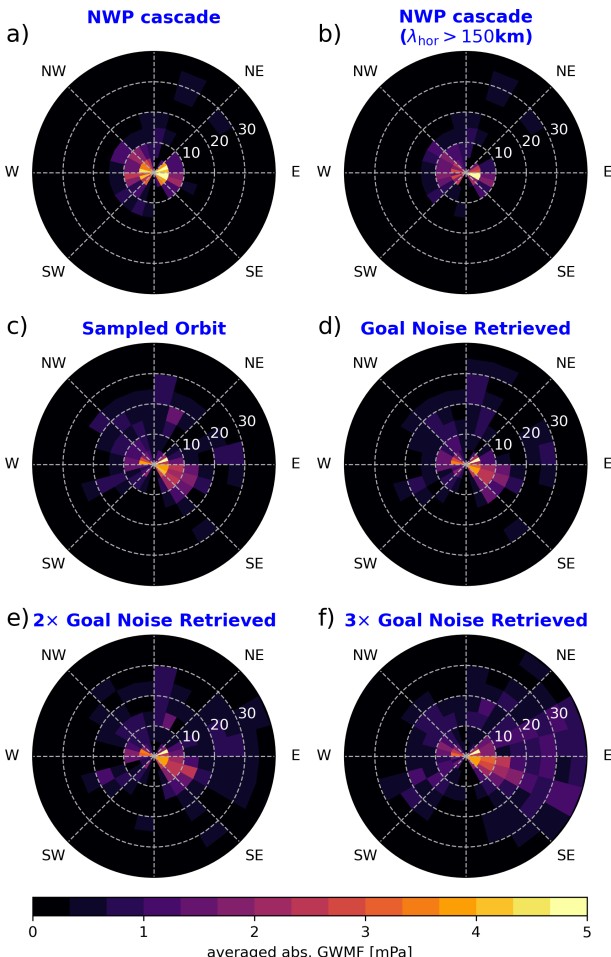

**Figure A2.** Same as Fig. A1 but for latitudes between $30°$S to $85°$N

faster waves (around $12.5\,\mathrm{ms}^{-1}$) in the full model. Considering both cases, we could say that the noise manifests as GWs with a phase speed around $15\,\mathrm{ms}^{-1}$.

## Appendix B: Assessment of JAGUAR simulations

For a quantification of the matching between the GWMF and GW drag estimated from JAGUAR directly from the wind
fluctuations on the one hand and S3D applied to retrieved temperatures on the other hand shown in Fig. 15, Fig. B1 shows the respective correlations across the full altitude range. In particular, the correlation of zonal mean GWMF is very high (0.92) up to about 65 km. Above this altitude, the retrieval deteriorates to some extend but the correlation still reaches about 0.8. Note

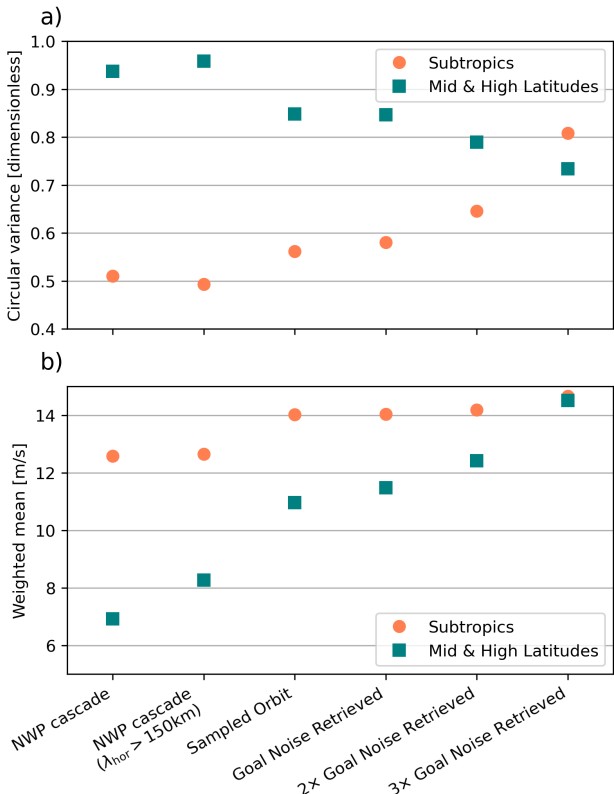

**Figure A3.** Circular variances and weighted mean phase speeds for the phase-speed spectra shown in Figs. A1 and A2. Subtropics are shown in orange circles, mid & high latitudes in teal squares.

that the correlation does not capture the bias between both data sets. The simulated CAIRT observations are fairly consistent at lower altitudes but are smaller than the model estimations at higher altitudes, e.g., at 65 km by about a factor of 1.5.

The GW drag shows lower correlations than the GWMF. Note, however, that the absolute GW drag is very small below about 55 km, and hence, smaller fluctuations lead to the lower correlations in the altitude range below. In particular, the patterns of alternating positive/negative drag in the altitude range between 40 km and 50 km (Fig. 16d and e) that is not seen in the JAGUAR data contributes to the low correlation.

    Figure B2 shows the scatter distribution of the zonal mean zonal GWMF and GW drag estimated from the JAGUAR winds 830 and CAIRT simulations, respectively. Note the different horizontal and vertical axes to compensate biases. The distributions are fairly straight up to the far edges. However, the CAIRT simulations underestimate the GWMF and the GW drag by about a factor of 1.5 and 2, respectively. This bias is mostly stemming from the highest altitudes, where the correlations deteriorated as well. This can be seen in the scatter distribution shown in Fig. B3 for the maps of GWMF at 70 km shown in Fig. 17, which shows a bias towards smaller GWMF in particular for regions where the GWMF in the model data is low as well. Stronger

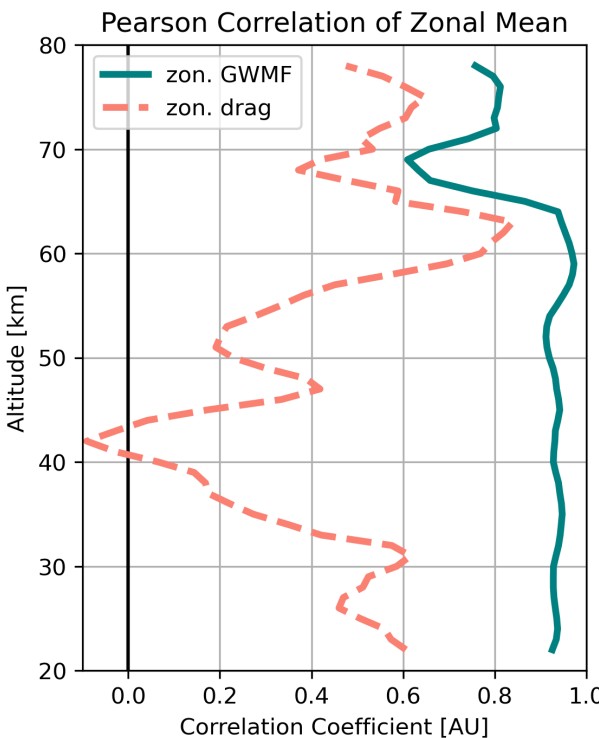

**Figure B1.** Correlation of GWMF (solid, teal) and GW drag (dashed, orange) between derivation from wind fluctuations and CAIRT simulations that are shown in Fig. 16 a and b and Fig. 16 c and d, respectively. Note that the absolute GW drag is small for altitudes below about 55 km.

GWMF events are with lower bias and can thus be trusted in the simulated observations. This is in agreement with seeing all the strong GW events in the horizontal distribution shown in Fig. 17.

*Author contributions.* SR and PP conceptualized the study, JU performed the synthetic retrievals for CAIRT, and IP provided the ECMWF high-resolution data. KS and SW provided the JAGUAR high altitude model data. SR and PP analyzed the data and wrote the paper with helpful discussions and input from all authors.

*Competing interests.* The authors declare that they have no conflicts of interest.

*Acknowledgements.* We would like to thank both anonymous reviewers for their constructive comments, which greatly helped in improving our article. The JAGUAR simulations were performed using the Earth Simulator at JAMSTEC.

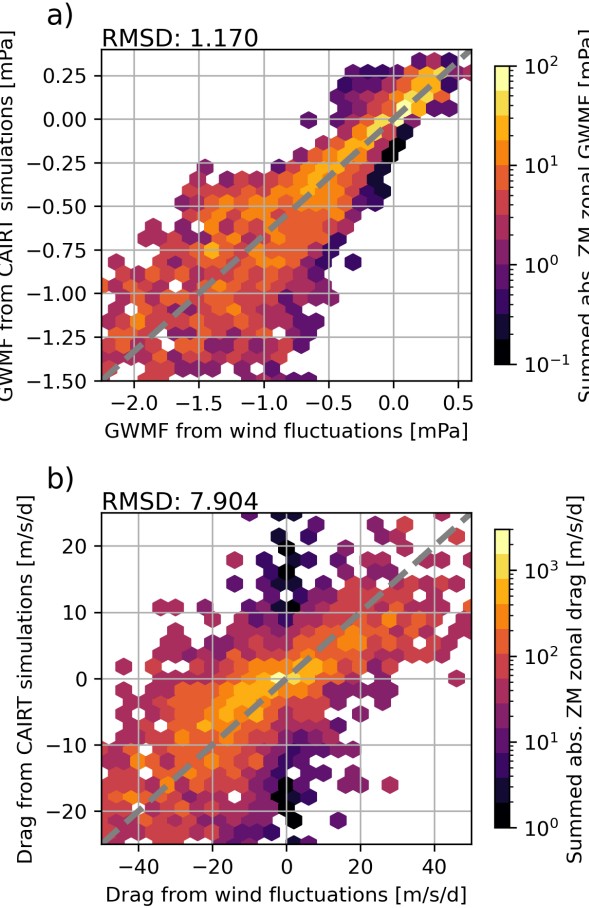

**Figure B2.** Scatter distribution comparing the zonal mean zonal GWMF and GW drag presented in Fig. 16a and b and Fig. 16c and d, respectively. Color shading gives the total absolute zonal GWMF or GW drag in the given bin. Note the different horizontal and vertical axes.

*Financial support.* This work was performed in support for CAIRT. ESA contributed to the funding via SciReC (Science Requirements Consolidation) and CEEPS (CAIRT End-to-End Performance Simulator) studies. Further, KS and SW were supported by JSPS KAKENHI Grant JP22H00169. SW was supported by the MEXT program for the advanced studies of climate change projection (SENTAN) Grant Number JPMXD0722681344.

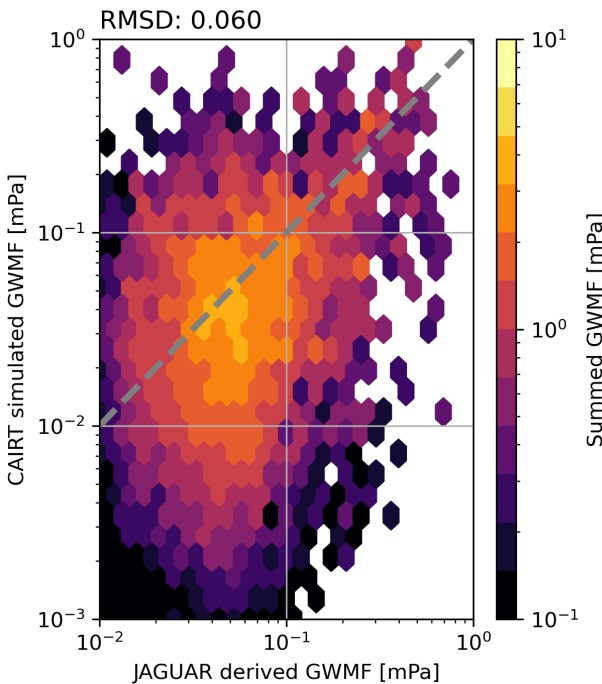

**Figure B3.** Scatter distribution of GWMF estimated from JAGUAR (horizontal axis) and CAIRT simulations (vertical axis) at 70 km shown in Fig. 17. Color shading shows the summed GWMF in the respective bin. Note the logarithmic scale.

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
