# Peer review of "Global scale gravity wave analysis methodology for the ESA Earth Explorer 11 candidate CAIRT"

_EGUsphere, 2024_

## Author Comment (AC1)

**1 Response Referee No. 1**

We thank the reviewer for the thorough review of our article. The comments and suggestions were very helpful in improving the presentation of our work and refining the science.

The responses to specific comments are given below. The original reviewer comments are given in italic and any text given in blue has been added to the manuscript in response to the comment.

**1.1 Overall comment**

*In this manuscript, the motivation for and the methodology of the atmospheric gravity wave research using The ESA Earth Explorer 11 candidate CAIRT is outlined and demonstrated. The prospective CAIRT mission would have an enormous impact on the atmospheric dynamics and transport research and for research of the middle atmosphere in general. Here, a group of world leading experts presents a cutting edge methodology that documents the ability of CAIRT to derive unprecedented information on the GW field and wave-mean flow interaction. The paper is clearly structured and very well written and I can recommend it for publication after the authors revise some technical issues and typos that I list below.*

*However, I feel that besides being just published, the paper has a potential to become a highlight paper and to have a really big impact on the community. But, sadly, I feel that in the current form the paper falls short of that, esp. in terms of readability of the text. Hence, I also give below some editorial comments/suggestions that should not prevent publication but if the authors pick them, they can possibly help improving the paper in this regard.*

**1.2 Specific comments**

1. *L25 - On breaking...*

   Rephrased to: "The GWs release this momentum when breaking,..."

2. *L30 ..GWs are of small horizontal scales compared to climate model resolutions and are, hence, not well taken into account for long-term projections..This is a low quality sentence. I think that the issue of underrepresentation of parts of GW spectra in climate and global weather prediction models deserves a more thorough discussion. Also, there should be a paragraph on how the unresolved effects are included (parameterizations). Clearly distinguishing between orography and non-orography GWs and highlighting how important the parameterizations are for climate models (for orography GWs - e.g. impact on SSW frequency Sigmond et al. (2023, GMD), Fig. 18); impact on stratospheric dynamics in general (Hajkova and Sacha, 2024, CliDyn) and for non-oro GW schemes e.g. Choi et al. (2018, Asia-Pacific J Atmos Sci)*

   The following paragraph has been added to address this point:

   "Even modern climate models are not able to resolve the full GW spectrum due to limited spatial resolution and hence the unresolved GWs are parameterized to approximate their effects. Non-orographic and orographic GWs are considered separately due to their distinct phase speed spectra and source processes. Both types are crucial for the performance of the climate model in long-term projections. For instance, they influence the frequency of SSW events [Sigmond et al., 2023, , Fig. 18] and the general dynamics of the stratosphere [Hájková and Šácha, 2024] for the orographic component and impact the QBO frequency [?Bushell et al., 2020, Richter et al., 2020] and forecasting performance [Choi et al., 2017, 2018, Kautz et al., 2020] for the non-orographic component. A better representation of these unresolved GWs creates the need for direct observations to further understand their role in atmospheric processes."

3. *L43..costly infrastructure.. the people working with ground based measurements on the other side argue that the ground based measurements are very cost effective compared to satellite obs. Can you substantiate your claim here?*

   Indeed this was laid out misleadingly. The sentence was changed to: "In addition, also remote sensing instruments have been deployed on research aircraft in dedicated GW campaigns [Fritts et al., 2016, Krisch et al., 2017, Rapp et al., 2021] and radiosondes are launched daily in a worldwide network."

4. *L84 ..intermittency - Here, again would be useful to distinguish between non-orographic and orographic parameterizations, where the issue of intermittency has been also discussed (e.g. Kuchar et al. (2020, WCD))*

   Good point, a sentence on this was added to the manuscript: "It is noteworthy that the intermittency is very different for orographic (high intermittency and seasonality) and non-orographic (comparatively low intermittency) GW parametrization schemes [Kuchar et al., 2020]."

5. *L89 ...propagate..here your discussion is inaccurate. GWs in the parameterizations (outside MS-GWaM) do not propagate at all. Only the saturation criterion is evaluated in a vertical column. Again, I argue that a paragraph devoted to importance and limitations of GW parameterizations is needed.*

   The corresponding text was changed to: "It became evident, though, that GW parametrizations neglect an important feature of GWs, i.e., that GWs in the real atmosphere spread not only vertically, as assumed in the parametrizations, but also laterally. In addition, no real vertical propagation is modeled (including group velocity and possible horizontal refraction) but a check of the saturation criteria within the column is performed. ..."

6. *L93..a major part...I would say that for current GCMs it would be more accurate to say some part of the GW spectrum (their effective resolutions are much worse than simply their horizontal resolution).*

   Indeed, the major is a bit of an overstatement considering GCMS. We changed the text as you suggest: "...resolving the meso- to long-scale part of the GW spectrum..."

7. *L99 the whole spectrum of GWs - again for accuraccy I would say almost the whole spectrum*

   Agree, changed as you suggest.

8. *L135 .. and that it is possible from the aircraft - please rewrite*

   The edited sentence now reads: "The studies demonstrate that high-accuracy wave determination is required for the scientific need and that airborne observations can deliver the necessary data of sufficient quality."

9. *L150 depends its*

   Changed to: "...the result depends on the configuration..."

10. *L159 .. , from where the SSW propagates downward.- This is certainly not true for the SSWs in general.*

    This subordinate clause has been removed.

11. *L161 For agreeing results - rewrite.*

    We changed the sentence to: "To achieve consistent results, the GW parameters must be known with high accuracy; therefore, a discussion on the impact of instrument noise on these spectra is presented."

12. *L165 GCM model data - rewrite*

    The sentence has been changed to: "This paper begins with an overview of the high-resolution GCM data and simulated CAIRT retrievals used for the assessment."

13. *L169 - here you say that you use NWP not GCM data..*

    Corrected in the new revision.

14. *L173 Sec .6*

    Corrected to "Sec. 6"

15. *L176 an estimate of an actual observation - an estimate of a potential observation?*

    Agree, this would be more precise. We changed the manuscript accordingly.

16. *L179 comparison to model data is based on model data - rephrase*

    This sentence was changed to: "The bulk of the simulated observations and following analysis , as well as the comparison to model data, is based on the Integrated Forecast System (IFS) of the European Centre for Medium-Range Weather Forecasts (ECMWF)."

17. *around L229 Is interpolation really the correct way and should not the spatial averaging be used to mimic what the coarse resolution instrument would see?*

    Indeed a spatial averaging using the correct kernel of the instrument would be the ideal way to treat the conversion from model to instrument data. The interpolation used here is an approximation but our assumption is that the along-track smoothing of the retrieval dominates.

    The corresponding section has been expanded with another note: "As an approximation, the model data is interpolated to the "retrieval" grid using a spline interpolation. Note that the resolution of the model data is finer than the observation sampling grid, and thus, the interpolation error will be small. A spline interpolation has been used because a simple linear interpolation tends to reduce GW amplitudes, which shall be avoided for a realistic comparison between simulated observations and original model data. Further note, that the retrieval will strongly smooth the temperature in along-track direction. The pixel width in across-track direction, however, is neglected."

18. *Tab. 1 - I would prefer additional column showing the meaning of the numbers (e.g. band, range or something). As it is now, you do not need a table.*

We incorporated the spectral nads into the main text and removed Tab. 1: "These spectral bands capture the strong emissions of three $CO_2$ Q-branches (719.0 cm$^{-1}$ – 721.0 cm$^{-1}$, 741.0 cm$^{-1}$ – 741.8 cm$^{-1}$, and 791.0 cm$^{-1}$ – 792.4 cm$^{-1}$) as well as an atmospheric window (831.0 cm$^{-1}$ – 832.0 cm$^{-1}$) to provide a background value."

We added a column of the relevant emitter in the respective spectral bands, i.e., whether it's a $CO_2$ Q-branch of the atmospheric window.

19. *L255 vertical flux of horizontal pseudomomentum - please state clearly that you are computing only a simplified version of the vertical flux of horizontal pseudomomentum valid under certain assumptions*

The following sentence was added to not the limitation and give the reference for more detail:

"...component, see Sec. 3.1). Note that this analysis is limited by the assumption that the background atmosphere allows for the WKB limit [Fritts and Alexander, 2003]. The..."

20. *L277 We all know that the Earth is not a sphere*

Yes, that's true, we meant to address the periodicity along the longitude. We changed the sentence as follows to better convey and clarify what we wanted to say:

Due to periodicity along a latitude circle, the zonal mean always covers a multiple of the wavelength of any wave in zonal direction, and hence, the zonal mean of the zonal momentum flux, $F_x$, can be used as a valid reference for the comparison.

21. *L277 ..comprises always a multiple of full wavelengths of any wave in zonal direction...I do not understand this, certainly not all wavelengths satisfy this.*

See comment above.

22. *L278 Fx, is a true reference ...I do not understand the meaning of this statement.*

See comment above.

23. *L280 is not a true reference...the same as above.*

Accordingly to the previous comment, the sentence was changed to: "...shows oscillations and can not be taken as a fully valid comparison reference."

24. *L281 ..For the closure of the momentum budget, the pseudo-momentum flux needs to be calculated.. - The momentum budget can certainly be closed without pseudo-momentum, only it takes a less elegant form. Also, you should be aware that you compute only a very simplified form of a pseudo-momentum (homogenous background winds..).*

Agree and, in particular, this sentence was not what we intended to convey. It is change in the revised manuscript to:

"For GWs, the pseudo-momentum flux is the quantity to consider for the interaction with the mean flow [Fritts and Alexander, 2003]."

25. *L297.. linked to the temperature amplitude (Ern et al., 2004) - again, include assumptions of this link (mid-frequency range etc.)*

The sentence has been changed to: "The GWMF values, which are directly linked to the temperature amplitudes [valid in the WKB limit, Ern et al., 2004], are therefore representative of the volume of the refit.

26. *L347 Chí squared was not defined before?*

This has been changed to: $\left(f(\vec{x}) - y(\vec{x})\right)^2 + P(\vec{k})$

In addition, the following sentence was added: "Here, $y(\vec{x})$ are the observed or modeled temperature values at the corresponding locations $\vec{x}$."

27. *L349-L356 and the whole subsection - There is so much of text and too few formulas or visuallization. It is really hard to follow the rationale behind the penalty in particular..*

Indeed, this section was not very helpful with the details on the penalty function. Therefore, the section has been rewritten as follows:

We choose the penalty function $P$ as a squared cosine function truncated at the first minimum:

$$P(\vec{k}) = \sum_{i=1}^{3} \begin{cases} 0 & |k_i| < w_i \\ h_i \cos^2\left(\frac{\pi}{2} \frac{|k_i| - k_{\mathrm{ny}}}{w_i}\right) & else \end{cases}, \tag{1}$$

where $k_i$ are the components of the wave vector $\vec{k}$ and $k_{\mathrm{ny}}$ is the Nyquist wave number, i.e., the highest resolvable wave number. The free parameters are the penalty heights, $h_i$, and widths, $w_i$, and the normalization is chosen such that the penalty smoothly transitions from $h_i$ at the Nyquist wave number to 0 at $|k_i| = k_{\mathrm{ny}} - w_i$. Throughout our analysis, the penalty widths $w_i$ are chosen as 0.8 times the spectral sampling in all three spatial dimensions. The penalty heights, $h_i$, need to be adapted in a way that most of the fitted noise is removed from the results. Still remaining solutions close to Nyquist limit are removed in the post processing.

28. *L367 ..local phenomena cancel in the zonal mean... for the zonal mean, GWDxx averages out by definition, but not the components connected with local meridional propagation*

    Indeed, we hope to clarify this by the following changes: "In general, it is expected that these local phenomena cancel for the zonal direction in the zonal mean and a general trend remains visible. However, even the zonal mean may show signs of lateral propagation in the meridional direction."

29. *L396 ..local events..-¿local peaks?*

    Agree, changed as suggested.

30. *Figure 4..panel a)... panel b)...*

    Corrected as suggested.

31. *Paragraphs around L445 and 450 - Please rewrite this section of results in a more consistent manner and do not jump between Fig. 5 and Fig. 6*

    We changed the corresponding part to:

    "Similarly, the eastward maximum in the southern subtropics and the westward maximum in the northern mid and high latitudes are predominantly composed of eastward and westward propagating waves, respectively. The east-west distribution will vary under different meteorological conditions. The zonal mean generated from the S3D data aligns very well with the general pattern and magnitude. Only at around 38°N, the S3D method underestimates the GWMF in comparison to the GWMF derived directly from the wind fluctuations. All in all, this shows that the S3D method is a valid tool for the estimation of GW distributions from model and satellite data.

    The good matching quality is also evident in the meridional GWMF (Fig. 6a), which is separated into northward and southward directed waves. The GWMF shows notable compensation between these components, as will be demonstrated more clearly in the phase speed spectra in Sec. 4.3."

32. *L489 In general, the good resemblance of the reference phase speed spectrum gives confidence that CAIRT would see a representative sample to estimate the full GW phase speed spectrum even in the case of higher-than-expected noise...Based on the figure, I am afraid that this is true only for the slow, likely orographic GWs. It seems like the non-orographic part (modes with large phase speeds) will be increasingly plagued by artificial signals. This can influence reliability of the ray tracing. Generally I think that it would be good to separate a little bit more between OGWs and non-OGWs in description of your results (I know that it is impossible to distinguish clearly between them).*

    To investigate the different effects of the retrieval on phase speed spectra for orographic and non-orographic GWs, we have added a section to the Appendix. There, we look at phase speed spectra for the subtropics as a proxy for non-orographic GWs and the mid to high latitudes for orographic GWs. Of course the separation is not perfectly valid but only an approximation.

    The section reads as follows:

    **Appendix A: Latitude-separated phase-speed spectra**

[revised manuscript text omitted]

33. *L504 - From Fig. 2, by eye, it seems that the biggest GWMF values are seen above Japan, but in Fig. 8 this is not that clear. Is it only due to subjective perception or is it likely a result of filtering of shorter waves?*

Both are a likely reason here. For one, we replaced Fig. 2 with the one displayed in Fig. R4. The white coastlines in combination with the chosen color map has increased the visibility of the region around Japan. And, as you mention, the horizontal wavelengths around Japan are fairly short in comparison (around 200 km).

34. *L531 ..GWs assume long vertical wavelengths..please rephrase*

This sentence has been rephrased to: "The background wind velocity is a crucial factor influencing the GWMF distribution because GWs tend to have long vertical wavelengths and can reach their largest amplitudes when propagating against strong background winds [Preusse et al., 2006]."

[Figure]

Figure R3: Circular variances and weighted mean phase speeds for the phase-speed spectra shown in Figs. R1 and R2. Subtropics are shown in orange circles, mid & high latitudes in teal squares.

35. *L576.. As a rule of thumb...rephrase and rather state the assumptions behind this estimate*

    This sentence has been changed to: "In mid-frequency approximation and in the stratosphere, where $N \approx 0.02\,\mathrm{s}^{-1}$, 3 km vertical wavelength corresponds to about $10\,\mathrm{m\,s}^{-1}$ (intrinsic) phase speed."

36. *L609.. panel a) ...panel b)*

    Changed as suggested.

37. *Figure 15.... Zonal mean zonal GWMF (a, b) and zonal mean zonal drag (c, d, e) deduced from JAGUAR model wind fluctuations (a, c), from CAIRT-sampled data (e), and from end-to-end simulations including radiative transfer, instrument noise and retrieval (b, d)...It is so difficult to understand this caption, please rewrite..*

    Indeed, the caption was confusing. We changed it to:

    "Zonal mean zonally directed GWMF derived from JAGUAR wind fluctuations (a) and from S3D analysis of the end-to-end simulated CAIRT temperature (b). Panels c)–e) show the zonal mean GW drag derived from the JAGUAR wind fluctuations (c), end-to-end simulated CAIRT temperature (d), and model data sampled to the CAIRT grid (e)."

38. *around L653 .... The comparison to the S3D analysis directly applied to the JAGUAR data (Fig. 17b and d) confirms the strong bias to the orbit tracks at 25 km. This bias is alleviated to some extent at 45 km altitude and the most prominent large-scale structures are resembled within the ray-tracing simulation. Also note that the ray tracing results are better at resembling larger scale structures simply due to more and stronger GWs being present in these regions (e.g., southern subtropics, Himalaya, and Scandinavia)....The discussed features are really hard to see and follow. It would be better to find a way of presenting this to get more quantitative.*

    Actually the JAGUAR data in the second column of Fig. 17 was very misleading as the shown data was the wrong altitude. This has been resolved by exchanging it for the following figure, where also the color scale has been extended:

    A further quantification of the ray tracing approach is planned for a follow-up study.

39. *Figure 18 and general - Opposed to tracing GWs to sources, I am not convinced about the utility of ray tracing for deriving GWD estimates. Are you not working outside of the underlying ray tracing*

[Figure]

**abs. GWMF — sum of wave components**

abs. GWMF [mPa]

Figure R4: GWMF estimated from the S3D wave analysis cascade with cuboid sizes given in Tab. 2 applied to the ECMWF IFS data. Shown is the sum of all four wave components at 25 km altitude for 3 January 2006 at 12:00 UTC.

*assumptions? On the other side, I acknowledge that the resulting zonal mean drag in Fig. 18 indeed looks more "realistic".*

Indeed, we are close to the limits of (linear) ray tracing. In particular, when GWs approach a critical level, the assumptions become invalid and the ray tracing breaks down. In our setup, however, we stop the ray tracing below a critical level before non-linear effects become a problem.

But of course, the approach using ray-tracing is still an approximation, albeit one that has been successfully tested with GW measurements by Krasauskas et al. [2023]. GW parameterizations use similar approximations of linearity and sub-sequentially derive the drag and we believe this is a better approximation, although still not 100% correct.

40. *L666... deviated as the vertical derivative...*

    "deviated" has been replaced by "calculated".

41. *L680... (e.g. ??))...*

    Ooops, this should have been Schoeberl and Newman [1995] and Hegglin et al. [2004]. Corrected in the revised version

42. *L689.. They inspired the development of global atmospheric models. -¿ I think that you want to say.. They inspired the development of GW parameterizations in global atmospheric models.*

    Indeed, we did not want to say that satellites have inspired GCMs as a whole. Changed as suggested.

43. *L710 ...(also, without penalty, the target range of wavelengths is not reliable; not shown)..maybe this should be shown at a relevant place of the manuscript, because I have found the penalty discussion hard to follow earlier in the text.*

    We hope that the updated discussion of the Nyquist penalty is more clear such that this part of the conclusion can be followed more easily. In addition, the corresponding sentence has been modified as follows:

    "For CAIRT retrievals, a Nyquist penalty has to be included, which rids the obtained wave spectrum of nonphysical and noise-like waves at the shortest theoretically resolvable scales (also, without penalty, the target range of wavelengths is not reliable; not shown)."

[Figure]

Figure R5: Absolute GWMF from ray tracing experiment. The rays were initialized from the S3D analyses data of 18th December 2018 00:00 UTC at 15, 20, 25, 30, and 35 km altitude from the JAGUAR simulations (see Sec. 5.3). Panels (a) and (b) show the resulting absolute GWMF at 45 km and 25 km, respectively.

**1.3    General comments**

44. *Figs. 8, 9, 15, 16..Instead of subjectively interpreting the information seen in the swaths, can you come with some more quantitative way of presenting these results? (less prone to subjective perception).*

For a quantification of the deterioration of the global maps in Figs. 8 and 9, we added the following paragraphs:

[revised manuscript text omitted]

45. *Footnotes - I do not consider it necessary that the footnotes are included*

    Thank you for the feedback. The footnotes are removed in the revised version.

46. *In addition to the analysis presented I would like to see it addressed in the manuscript, whether the CAIRT observations can contribute with information on short-term variability of GWD (hours, days?) and also long-term trends (Is the projected accuracy and stability of the measurements sufficient for this?)*

    The CAIRT mission is planned to be at least a 5-year mission and with fuel and support for up to 10 years. Trends visible in this period could be observed, e.g., tropical GW activity. For high latitude trends, the number of observed seasons could be a limiting factor. Currently there is no reason to believe that the instrument will degrade during its deployment. The following paragraph has been added to the conclusions:

    Beyond individual observations, the CAIRT mission might be able to estimate trends in GW activity during it's planned lifetime of 5 years (with a threshold lifetime of up to 10 years) but this is beyond the current study. The daily global coverage allows for observation of daily variability of the global GW distributions.

**References**

A. C. Bushell, J. A. Anstey, N. Butchart, Y. Kawatani, S. M. Osprey, J. H. Richter, F. Serva, P. Braesicke, C. Cagnazzo, C.-C. Chen, H.-Y. Chun, R. R. Garcia, L. J. Gray, K. Hamilton, T. Kerzenmacher, Y.-H. Kim, F. Lott, C. McLandress, H. Naoe, J. Scinocca, A. K. Smith, T. N. Stockdale, S. Versick, S. Watanabe, K. Yoshida, and S. Yukimoto. Evaluation of the Quasi-Biennial Oscillation in global climate models for the

[Figure]

Figure R7: Correlation of GWMF (solid, teal) and GW drag (dashed, orange) between derivation from wind fluctuations and CAIRT simulations that are shown in Fig. 15 a and b and Fig. 15 c and d, respectively. Note that the absolute GW drag is small for altitudes below about 55 km.

SPARC QBO-initiative. *Quart. J. Roy. Meteorol. Soc.*, n/a(n/a), 2020. doi: https://doi.org/10.1002/qj.3765. URL https://rmets.onlinelibrary.wiley.com/doi/abs/10.1002/qj.3765.

H.-J. Choi, S.-J. Choi, M.-S. Koo, J.-E. Kim, Y. C. Kwon, and S.-Y. Hong. Effects of parameterized orographic drag on weather forecasting and simulated climatology over east asia during boreal summer. *J. Geophys. Res. Atmos.*, 122(20):10669–10678, OCT 27 2017. ISSN 2169-897X. doi: 10.1002/2017JD026696.

H.-J. Choi, J.-Y. Han, M.-S. Koo, Y.-H. Chun, Hye-Yeong Kim, and S.-Y. Hong. Effects of non-orographic gravity wave drag on seasonal and medium-range predictions in a global forecast model. *Asia-Pac. J. Atmos. Sci.*, 54(1):385–402, 2018. doi: 10.1007/s13143-018-0023-1.

M. Ern, P. Preusse, M. J. Alexander, and C. D. Warner. Absolute values of gravity wave momentum flux derived from satellite data. *J. Geophys. Res. Atmos.*, 109(D20), 2004. ISSN 2156-2202. doi: 10.1029/2004JD004752.

D. Fritts and M. Alexander. Gravity wave dynamics and effects in the middle atmosphere. *Rev. Geophys.*, 41 (1), APR 16 2003. ISSN 8755-1209. doi: 10.1029/2001RG000106.

D. C. Fritts, R. B. Smith, M. J. Taylor, J. D. Doyle, S. D. Eckermann, A. Doernbrack, M. Rapp, B. P. Williams, P. D. Pautet, K. Bossert, N. R. Criddle, C. A. Reynolds, P. A. Reinecke, M. Uddstrom, M. J. Revell, R. Turner, B. Kaifler, J. S. Wagner, T. Mixa, C. G. Kruse, A. D. Nugent, C. D. Watson, S. Gisinger, S. M. Smith, R. S. Lieberman, B. Laughman, J. J. Moore, W. O. Brown, J. A. Haggerty, A. Rockwell, G. J. Stossmeister, S. F. Williams, G. Hernandez, D. J. Murphy, A. R. Klekociuk, I. M. Reid, and J. Ma. The deep propagating gravity wave experiment (DEEPWAVE): An airborne and ground-based exploration of gravity wave propagation and effects from their sources throughout the lower and middle atmosphere. *Bull. Amer. Meteor. Soc.*, 97(3):425–453, MAR 2016. ISSN 0003-0007. doi: 10.1175/BAMS-D-14-00269.1.

D. Hájková and P. Šácha. Parameterized orographic gravity wave drag and dynamical effects in CMIP6 models. *Clim. Dyn.*, 62(3):2259–2284, 2024. doi: 10.1007/s00382-023-07021-0.

M. I. Hegglin, D. Brunner, H. Wernli, C. Schwierz, O. Martius, P. Hoor, H. Fischer, U. Parchatka, N. Spelten, C. Schiller, M. Krebsbach, U. Weers, J. Staehelin, and T. Peter. Tracing troposphere-to-stratosphere transport above a mid-latitude deep convective system. *Atmos. Chem. Phys.*, 4(3):741–756, 2004. doi: 10.5194/acp-4-741-2004. URL https://acp.copernicus.org/articles/4/741/2004/.

[Figure]

Figure R8: Scatter distribution comparing the zonal mean zonal GWMF and GW drag presented in Fig.15a and b and Fig. 15c and d, respectively. Color shading gives the total absolute zonal GWMF or GW drag in the given bin. Note the different horizontal and vertical axes.

L.-A. Kautz, I. Polichtchouk, T. Birner, H. Garny, and J. G. Pinto. Enhanced extended-range predictability of the 2018 late-winter Eurasian cold spell due to the stratosphere. *Quart. J. Roy. Meteorol. Soc.*, 146(727, B): 1040–1055, JAN 2020. ISSN 0035-9009. doi: 10.1002/qj.3724.

L. Krasauskas, B. Kaifler, S. Rhode, J. Ungermann, W. Woiwode, and P. Preusse. Oblique propagation and refraction of gravity waves over the Andes observed by GLORIA and ALIMA during the SouthTRAC campaign. *J. Geophys. Res. Atmos.*, page e2022JD037798, 2023. doi: 10.1029/2022JD037798.

I. Krisch, P. Preusse, J. Ungermann, A. Dörnbrack, S. D. Eckermann, M. Ern, F. Friedl-Vallon, M. Kaufmann, H. Oelhaf, M. Rapp, C. Strube, and M. Riese. First tomographic observations of gravity waves by the infrared limb imager GLORIA. *Atmos. Chem. Phys.*, 17(24):14937–14953, 2017. doi: 10.5194/acp-17-14937-2017.

A. Kuchar, P. Sacha, R. Eichinger, C. Jacobi, P. Pisoft, and H. E. Rieder. On the intermittency of orographic gravity wave hotspots and its importance for middle atmosphere dynamics. *Weather Clim. Dynam.*, 1(2): 481–495, 2020. doi: 10.5194/wcd-1-481-2020. URL https://wcd.copernicus.org/articles/1/481/2020/.

P. Preusse, M. Ern, S. D. Eckermann, C. D. Warner, R. H. Picard, P. Knieling, M. Krebsbach, J. M. Russell III, M. G. Mlynczak, C. J. Mertens, and M. Riese. Tropopause to mesopause gravity waves in August: measurement and modeling. *J. Atm. Sol.-Terr. Phys.*, 68:1730–1751, 2006. doi: 10.1016/j.jastp.2005.10.019.

M. Rapp, B. Kaifler, A. Dörnbrack, S. Gisinger, T. Mixa, R. Reichert, N. Kaifler, S. Knobloch, R. Eckert, N. Wildmann, A. Giez, L. Krasauskas, P. Preusse, M. Geldenhuys, M. Riese, W. Woiwode, F. Friedl-Vallon, B.-M. Sinnhuber, A. de la Torre, P. Alexander, J. L. Hormaechea, D. Janches, M. Garhammer, J. L. Chau, J. F. Conte, P. Hoor, and A. Engel. SOUTHTRAC-GW: An airborne field campaign to explore gravity wave dynamics at the world's strongest hotspot. *Bull. Amer. Meteor. Soc.*, 102(4):E871 – E893, 2021. doi: 10.1175/BAMS-D-20-0034.1.

J. H. Richter, N. Butchart, Y. Kawatani, A. C. Bushell, L. Holt, F. Serva, J. Anstey, I. R. Simpson, S. Osprey, K. Hamilton, P. Braesicke, C. Cagnazzo, C.-C. Chen, R. R. Garcia, L. J. Gray, T. Kerzenmacher,

[Figure]

Figure R9: Scatter distribution of GWMF estimated from JAGUAR (horizontal axis) and CAIRT simulations (vertical axis) at 70 km shown in Fig. 16. Color shading shows the summed GWMF in the respective bin. Note the logarithmic scale.

F. Lott, C. McLandress, H. Naoe, J. Scinocca, T. N. Stockdale, S. Versick, S. Watanabe, K. Yoshida, and S. Yukimoto. Response of the quasi-biennial oscillation to a warming climate in global climate models. *Quart. J. Roy. Meteorol. Soc.*, n/a(n/a), 2020. doi: https://doi.org/10.1002/qj.3749. URL `https://rmets.onlinelibrary.wiley.com/doi/abs/10.1002/qj.3749`.

M. R. Schoeberl and P. A. Newman. A multiple-level trajectory analysis of vortex filaments. *J. Geophys. Res. Atmos.*, 100(D12):25801–25815, 1995. doi: https://doi.org/10.1029/95JD02414. URL `https://agupubs.onlinelibrary.wiley.com/doi/abs/10.1029/95JD02414`.

M. Sigmond, J. Anstey, V. Arora, R. Digby, N. Gillett, V. Kharin, W. Merryfield, C. Reader, J. Scinocca, N. Swart, J. Virgin, C. Abraham, J. Cole, N. Lambert, W.-S. Lee, Y. Liang, E. Malinina, L. Rieger, K. von Salzen, C. Seiler, C. Seinen, A. Shao, R. Sospedra-Alfonso, L. Wang, and D. Yang. Improvements in the Canadian Earth System Model (CanESM) through systematic model analysis: CanESM5.0 and CanESM5.1. *Geosci. Model Dev.*, 16(22):6553–6591, NOV 15 2023. ISSN 1991-959X. doi: 10.5194/gmd-16-6553-2023.

---

## Author Comment (AC2)

**1 Response Referee No. 2**

We thank the reviewer for the positive review of our article. The comments and suggestions were very helpful in improving the presentation of our work and refining the science.

The responses to specific comments are given below. The original reviewer comments are given in italic and any text given in blue has been added to the manuscript in response to the comment.

**1.1 Overall comment**

*This paper constitutes a proof of concept for the ESA Earth Explorer 11 candidate CAIRT in terms of ability to provide valuable global information on gravity wave characteristics, including dissipation, throughout the middle atmosphere. Two high resolutions models are sampled to generate synthetical observations, mimicking the observational design of an the instrument on board (an limb imaging Michelson interferometer). The results show the great potential of the instrument.*

*The manuscript is clearly written, I only have a few minor comments, as listed below.*

**1.2 Specific comments**

1. *- Line 23-25. Although there exists a wave-induced flux of momentum, strictly speaking waves in fluids do not have momentum (McIntyre 1981). So (gravity) waves do not gain or release momentum. Perhaps it would be more precise to say that gravity waves transport (or redistribute?) momentum/energy within different layers of the atmosphere.*

   *McIntyre ME. On the 'wave momentum' myth. Journal of Fluid Mechanics. 1981;106:331-347. doi:10.1017/S0022112081001626 .*

   Agree, this was misleading. The sentence was changed to:

   From their excitation processes like flow over orography, convection, jet instabilities, and other effects [Fritts and Alexander, 2003], they carry momentum to higher layers of the atmosphere by wave propagation.

2. *- Lines 29-30. Radiative cooling is a very important driver of the vortex recovery after SSWs. Are GWs really the main driving force?*

   Indeed this is not true as it was written and thus the sentence has been changed accordingly. GWs contribute to the recovery of the stratospheric vortex and the downward propagation of the elevated stratopause. The updated text reads:

   "They are likely a major driving force in the recovery phase of the stratospheric vortex and the downward propagation of an elevated stratopause [Ern et al., 2016, Thurairajah and Cullens, 2022, Harvey et al., 2022, 2023]."

3. *- Line 45. This sentence is not well written, needs to be rephrased.*

   The sentence has been rephrased to:

   "Satellite missions are best suited for the long-term observation of large-scale momentum transport needed for understanding global-scale processes."

4. *- Lines 60-65. To my knowledge, the first successful implementation of a GW drag parameterization in a climate model (actually a NWP model) was reported by Palmer et al. (1986), not by Lindzen (1981). Lindzen's study was indeed one of the first to show that a parametrization based on linear saturation theory (convective instability) of a monochromatic wave could provide a dynamical forcing in the mesosphere able to explain the warm mesopause in winter and the cold in summer, but no GCM was used.*

   *Palmer, T.N., Shutts, G.J. and Swinbank, R. (1986), Alleviation of a systematic westerly bias in general circulation and numerical weather prediction models through an orographic gravity wave drag parametrization. Q.J.R. Meteorol. Soc., 112: 1001-1039. https://doi.org/10.1002/qj.49711247406*

   Yes, you are absolutely correct. The sentence was rephrased to:

   "However, the general idea of GWs accelerating large-scale winds, developed by Lindzen [1981] and first implemented in a GCM by Palmer et al. [1986], has proven to be essential for global wind-systems also in other parts of the atmosphere."

5. *- Line 187. At what altitude does the sponge layer start?*

   The sponge layer starts above 0.78 hPa. This information is included in the revised text:

"The simulations have 137 vertical levels including a sponge layer of reduced strength compared to the standard IFS setup in order to limit the damping of GWs in the upper levels. The sponge layer starts above $0.78\,\text{hPa}$ (about $50\,\text{km}$)."

6. - *Lines 360, 363. Vertical gradient of the GWMF → vertical derivative (or divergence of the momentum vertical flux).*

   Changed as suggested.

7. - *Line 368. They should cancel out also in the time mean, which may be more relevant if the process we are dealing with is wave propagation imprint on the EP flux divergence.*

   Indeed, uniformly distributed fluctuations would also cancel out in the time mean, while consistently unidirectional propagation would become visible. Since CAIRT has a long revisit time (compared to model time resolution), a time mean would only be possible for persistent GW activity in the monthly mean. This is of course something that would be very interesting to look at.

   By taking the zonal mean, we try to keep the observation period short, i.e., daily, while getting rid of propagation fluctuations in the zonal direction. Of course, consistent meridional GW propagation could still lead to spurious drag as discussed in the text.

   Surely they also cancel out in a time mean, however, we can not perform such an time averaging with CAIRT - only possible for longer term propagation studies, e.g., on a monthly mean basis.

8. - *Line 371. What is the meaning of "potential drag"?*

   The sentence has been extended for a clarification:

   "Regions where GWs propagate into are characterized by negative "potential drag" derived from absolute values of GWMF [Ern et al., 2011], i.e., GWMF increasing with altitude in a strict columnar consideration - something which should not occur according to classical theory."

9. - *Line 576-577. It would seem that those numbers have been calculated using the mid-frequency approximation of the wave dispersion relation. Please specify.*

   This has been changed for clarification and now reads as follows:

   "In mid-frequency approximation and in the stratosphere, where $N \approx 0.02\,\text{s}^{-1}$, $3\,\text{km}$ vertical wavelength corresponds to about $10\,\text{ms}^{-1}$ (intrinsic) phase speed."

10. - *Line 695. Many scientific articles that present results on observations of gravity waves, both from a global perspective or based on case studies, use the argument that GW parameterizations are poorly constrained by observations, and hence the need for those kind of studies. The argument is valid, but it is not straightforward how to use the information provided by observations to improve the calibration of the parameterizations. Perhaps the main reason for this is that the tunable parameters have to do with GW characteristics (e.g. momentum flux, phase speed spectrum, etc.) at the source level. But the propagation and dissipation of the waves are not easily tunable.*

    *One of the strengths of CAIRT in this respect is to be able to provide information on the dissipation of waves (phase speeds, drag, range of altitudes, etc.). This would be extremely valuable to assess the output of current parameterizations, study and refine their performance, and evaluate whether their level of complexity (WKB solutions, columnar approach, etc.) is good enough to emulate the observations.*

    Thank you for this comment and I think you might be right that typical parametrizatons follow sort of a "fire and forget" approach where most of the tuning happens by modulating the source spectra. And indeed CAIRT could provide further insights into the dissipation or even the importance of non-linear processes.

    A note on this has been added in the conclusions:

    "... Further, GW parametrizations will be in future use for long-term runs and require tuning to observations. And since CAIRT observes a wide range of altitudes, the GW dissipation and propagation are observed and could be used for more advanced parametrizations of these processes. The validity of the existing parametrizations could be investigated giving an estimation of the importance of non-linear effects. ..."

**References**

M. Ern, P. Preusse, J. C. Gille, C. L. Hepplewhite, M. G. Mlynczak, J. M. Russell III, and M. Riese. Implications for atmospheric dynamics derived from global observations of gravity wave momentum flux in stratosphere and mesosphere. *J. Geophys. Res.*, 116, 2011. doi: 10.1029/2011JD015821.

M. Ern, Q. T. Trinh, M. Kaufmann, I. Krisch, P. Preusse, J. Ungermann, Y. Zhu, J. C. Gille, M. G. Mlynczak, J. M. Russell, III, M. J. Schwartz, and M. Riese. Satellite observations of middle atmosphere gravity wave absolute momentum flux and of its vertical gradient during recent stratospheric warmings. *Atmos. Chem. Phys.*, 16(15):9983–10019, AUG 9 2016. ISSN 1680-7316. doi: 10.5194/acp-16-9983-2016.

D. Fritts and M. Alexander. Gravity wave dynamics and effects in the middle atmosphere. *Rev. Geophys.*, 41 (1), APR 16 2003. ISSN 8755-1209. doi: 10.1029/2001RG000106.

V. L. Harvey, N. Pedatella, E. Becker, and C. Randall. Evaluation of polar winter mesopause wind in WAC-CMX+DART. *J. Geophys. Res. Atmos.*, 127(15):e2022JD037063, 2022. doi: 10.1029/2022JD037063. URL https://agupubs.onlinelibrary.wiley.com/doi/abs/10.1029/2022JD037063.

V. L. Harvey, C. E. Randall, L. P. Goncharenko, E. Becker, J. M. Forbes, J. Carstens, S. Xu, J. A. France, S. r. Zhang, and S. M. Bailey. CIPS observations of gravity wave activity at the edge of the polar vortices and coupling to the ionosphere. *J. Geophys. Res. Atmos.*, 128(12), JUN 27 2023. ISSN 2169-897X. doi: 10.1029/2023JD038827.

R. S. Lindzen. Turbulence and stress due to gravity wave and tidal breakdown. *J. Geophys. Res.*, 86:9707–9714, 1981.

T. N. Palmer, G. J. Shutts, and R. Swinbank. Alleviation of a systematic weterly bias in general circulation and numerical weather prediction models trough an orographic gravity wave drag parameterization. *Quart. J. Roy. Meteorol. Soc.*, 112:1001–1093, 1986.

B. Thurairajah and C. Y. Cullens. On the downward progression of stratospheric temperature anomalies using long-term SABER observations. *J. Geophys. Res. Atmos.*, 127(11), JUN 16 2022. ISSN 2169-897X. doi: 10.1029/2022JD036487.